# Semiochemical responsive olfactory sensory neurons are sexually dimorphic and plastic

**Aashutosh Vihani[1]\*, Xiaoyang Serene Hu[2], Sivaji Gundala[3], Sachiko Koyama[4], Eric Block[3], Hiroaki Matsunami[1,2,5]\***

[1]Department of Neurobiology, Neurobiology Graduate Program, Duke University Medical Center, Durham, United States; [2]Department of Molecular Genetics and Microbiology, Duke University Medical Center, Durham, United States; [3]Department of Chemistry, University at Albany, State University of New York, Albany, United States; [4]School of Medicine, Medical Sciences, Indiana University, Bloomington, United States; [5]Duke Institute for Brain Sciences, Duke University, Durham, United States

**Abstract** Understanding how genes and experience work in concert to generate phenotypic variability will provide a better understanding of individuality. Here, we considered this in the main olfactory epithelium, a chemosensory structure with over a thousand distinct cell types in mice. We identified a subpopulation of olfactory sensory neurons, defined by receptor expression, whose abundances were sexually dimorphic. This subpopulation of olfactory sensory neurons was over-represented in sex-separated mice and robustly responsive to sex-specific semiochemicals. Sex-combined housing led to an attenuation of the dimorphic representations. Single-cell sequencing analysis revealed an axis of activity-dependent gene expression amongst a subset of the dimorphic OSN populations. Finally, the pro-apoptotic gene *Bax* was necessary to generate the dimorphic representations. Altogether, our results suggest a role of experience and activity in influencing homeostatic mechanisms to generate a robust sexually dimorphic phenotype in the main olfactory epithelium.

**\*For correspondence:**
aashutosh.vihani@duke.edu (AV);
hiroaki.matsunami@duke.edu
(HM)

**Competing interest:** See
page 24

**Reviewing editor:** Tali Kimchi,
Weizmann Institute of Science,
Israel

## Introduction

Emergence of individual variability is a ubiquitous feature across animals; individual variability refers to the differing biological factors ("nature"), experiences ("nurture"), and randomness (e.g., stochastic gene expression) that generate phenotypic variability. Evidence for this variability has led to extensive research on the adaptive significance and ecological, or evolutionary, consequences of individuality (*Torquet et al., 2018*). Nonetheless, insight into the relative contributions and proximal mechanisms of "nature" versus "nurture" in generating phenotypic variabilities have been largely elusive.

One robust example of nature-induced inter-individual variability is sexual dimorphism. Previous work has identified both anatomical and functional substrates of this variability in the nervous system of mice (*Shah et al., 2004*; *Yang et al., 2013*; *Li et al., 2017*; *Remedios et al., 2017*). Similarly, extensive literature points toward an essential role of nurture-induced variability by experience-dependent, or activity-dependent, neural plasticity (*Zhao and Reed, 2001*; *Zou et al., 2004*; *Holtmaat and Svoboda, 2009*). Here, to investigate how nature and nurture work in concert to generate phenotypic variability, we focused on the mouse main olfactory epithelium (MOE), a chemosensory structure devoted to the detection of volatile odor cues. Olfactory sensory neurons (OSNs) found in the MOE express a single olfactory receptor (OR) in a monoallelic fashion out of a large and

diverse family of over 1000 candidate genes (*Buck and Axel, 1991*; *Zhang and Firestein, 2002*), thus enabling this system with an incredible potential for the emergence of individual variability at the level of cell types. Odor recognition at the level of OSNs additionally has been shown to follow a combinatorial coding scheme where one OR can be activated by a set of odorants and one odorant can activate a combination of ORs (*Malnic et al., 1999*; *Jiang et al., 2015*). Through such combinatorial coding, it has been postulated that organisms, including mice and humans, can detect and discriminate against a myriad of odor molecules.

We performed RNA-Seq on the whole olfactory mucosa of mice of different sexes ("nature") and experiences ("nurture") to investigate origins of inter-individual differences. In doing so, we uncovered a subset of ORs that exhibit sexually dimorphic expression under sex-separated conditions. In situ mRNA hybridization probing for the expression of these ORs demonstrated the proportions of OSNs expressing these ORs to be differentially represented between the sexes. Activity-dependent labeling experiments further identified the female enriched subpopulation of OSNs as selective responders to odor cues generated by mature male mice. Targeted screening of previously identified sex-specific and sex-enriched volatiles demonstrated that this subpopulation of OSNs also responded robustly to the reproductive behavior and physiology modifying semiochemicals 2-*sec*-butyl-4,5-dihydrothiazole (SBT) and (methylthio)methanethiol (MTMT) in vivo. Similarly, male enriched subpopulations of OSNs were identified as selective responders to macrocyclic musk odorants. To test the role of experience in generating the sexually dimorphic representation of ORs, we switched male and female mice from sex-separated conditions to sex-combined conditions and learned that the dimorphic representations had severely attenuated. Examination of single-cell transcriptomes from male OSNs expressing sexually dimorphic ORs revealed an enrichment of activity-dependent gene expression amongst OSNs expressing ORs enriched in female mice. Finally, testing of sex-separated mutant mice null for the pro-apoptotic gene Bcl2-associated X protein (*Bax*$^{-/-}$) revealed a failure to generate robust sexual dimorphisms within the whole olfactory mucosa. Altogether, our results suggest a link between specific olfactory experiences and OSN lifespan via activity as a means to influence sensory cell-level odor representations in the olfactory system.

## Results

### Identification of sexually dimorphic ORs in the MOE

We first performed RNA-Seq on the whole olfactory mucosa of male and female mice at various ages housed under sex-separated conditions after weaning (*Figure 1A*). Differential expression analysis of ORs revealed no obvious sexually dimorphic OR expression at 3 weeks (weaning) age. In contrast, progressive differential expression analysis of ORs at 9, 26, and 43 weeks age revealed at least three OR genes, *Olfr910*, *Olfr912*, and *Olfr1295*, to exhibit growing and robust enrichment in female mice (male enriched ORs are discussed later; *Figure 1B–E*). Examination of the proportion reads aligned to each of these ORs longitudinally revealed amplification of the dimorphism between the sexes with longer-term sex-separation (*Figure 1F*). Furthermore, this amplification appeared to exhibit receptor-specific patterns, as *Olfr1295* exhibited near-maximally dimorphic enrichment in female mice by 9 weeks age. In contrast, *Olfr910* and *Olfr912*, which did not exhibit obvious dimorphic expression at 9 weeks age, were robustly dimorphic by 43 weeks age (at 43 weeks old: *Olfr910* log$_2$FC = 2.19 FDR < 6.71E-26, *Olfr912* log$_2$FC = 1.97 FDR < 3.41E-14, *Olfr1295* log$_2$FC = 2.55 FDR < 5.83E-19).

Past work demonstrating a correlation between OR transcript abundance and the number of OSNs expressing those ORs led us to quantify the number of OSNs expressing these ORs (*Khan et al., 2011*; *Ibarra-Soria et al., 2017*). We probed specifically for the expression of *Olfr910*, *Olfr912*, and *Olfr1295* on the MOE of sex-separated male and female mice using anti-sense probes against the 3′ untranslated regions (UTRs) of each of these ORs (*Figure 2—figure supplement 1A-C*). The results of the in situ mRNA hybridization demonstrated that the proportion of OSNs expressing *Olfr910*, *Olfr912, and Olfr1295* was greater in 43-week-old female mice than in 43-week-old male mice (all p < 0.0001, unpaired two-tailed t-test; *Figure 2A–C*). Altogether, these results demonstrate that the subpopulation of OSNs expressing *Olfr910*, *Olfr912*, and *Olfr1295* are over-represented in sex-separated female mice compared to sex-separated male mice.

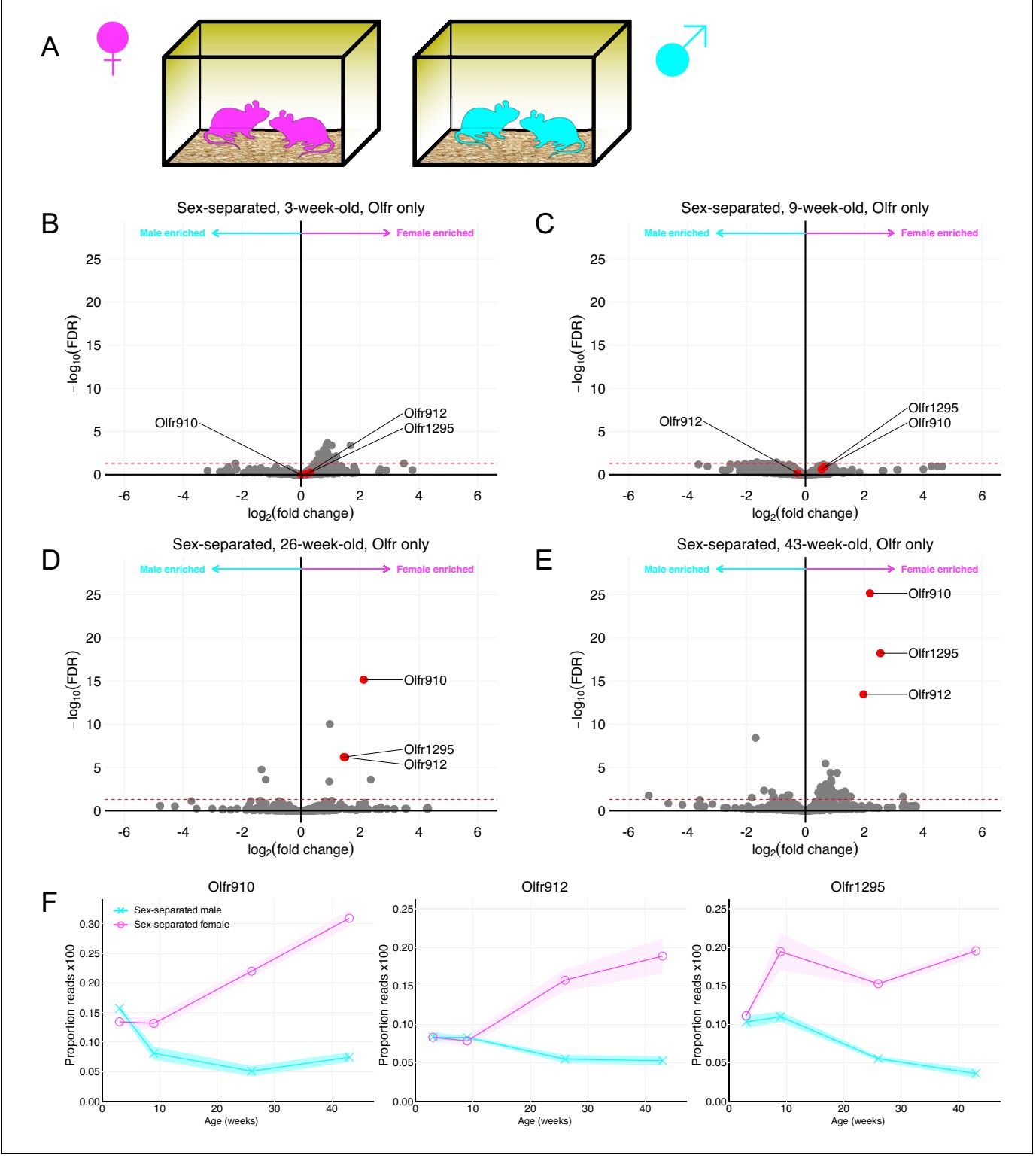

**Figure 1.** Identification of ORs exhibiting sexually dimorphic expression in sex-separated mice. (A) Schematic of the housing setup. For sex-separation, male mice were co-housed exclusively with male mice. Female mice were co-housed exclusively with female mice. (B) Volcano plot comparing expression of *Olfrs* between 3-week-old sex-separated male and female mice. *Olfr910*, *Olfr912*, and *Olfr1295* are highlighted in red. The red dashed line indicates an FDR = 0.05. Data are from n = 3 male and n = 3 female mice. (C) Volcano plot comparing expression of *Olfrs* between 9-week-old sex-separated male and female mice. (D) Volcano plot comparing expression of *Olfrs* between 26-week-old sex-separated male and female mice. (E)
*Figure 1 continued on next page*

*Figure 1 continued*

Volcano plot comparing expression of *Olfrs* between 43-week-old sex-separated male and female mice. (F) Longitudinal plotting of the mean and SEM of proportions of reads aligned to *Olfr910*, *Olfr912*, and *Olfr1295* in sex-separated male and female mice.

## Sexually dimorphic OSNs are selectively activated by the scent of adult male mice in vivo

Our identification of a subpopulation of OSNs over-represented in sex-separated female mice led us to hypothesize a role of the associated ORs in detecting sex-specific odors. To test this hypothesis, we briefly exposed juvenile mice (~3 weeks old) to either a blank odor cassette (control), mature male mice, mature female mice, or 10 µL of 1% (v/v) acetophenone spotted onto blotting paper placed inside an odor cassette (*Figure 3A*). We specifically focused on OSNs expressing *Olfr910*, *Olfr912*, or *Olfr1295* by in situ mRNA hybridization and performed immunostaining for the phosphorylation of ribosomal subunit S6 (pS6), a pan-neuronal marker of activity (*Knight et al., 2012*; *Jiang et al., 2015*).

Double in situ mRNA hybridization and immunostaining revealed the population of OSNs expressing *Olfr910*, *Olfr912*, and *Olfr1295* to display elevated pS6 staining intensity only when exposed to mature male mice (all p < 0.0001, one-way ANOVA with Dunnett's multiple comparisons test correction; *Figure 3B–D*, *Figure 3—figure supplement 1A-C*). Exposure to mature female mice or acetophenone did not lead to significant pS6 induction in the population of OSNs expressing either *Olfr910*, *Olfr912*, or *Olfr1295* (all p > 0.05, one-way ANOVA with Dunnett's multiple comparisons test correction; *Figure 3B–D*, *Figure 3—figure supplement 1A-C*). Altogether, these results demonstrate that the subpopulation of OSNs that express *Olfr910*, *Olfr912*, or *Olfr1295* to be responders to the natural scent of mature male mice.

## Sexually dimorphic OSNs are selectively responsive to specific semiochemicals in vivo

The observation that OSNs expressing *Olfr910*, *Olfr912*, and *Olfr1295* are activated by the scent of mature male mice led us to hypothesize that this subpopulation of OSNs responds specifically to sexually dimorphic odors produced by mature male mice. To test this hypothesis, we searched the literature for known sex-specific or sex-enriched odors and leveraged phosphorylated S6 ribosomal subunit capture (pS6-IP) as a means to determine the molecular identities of the OSNs activated by monomolecular odorants in vivo. Immunoprecipitation of phosphorylated ribosomes from activated neurons, followed by associated mRNA profiling by RNA-Seq (pS6-IP-Seq) and differential expression analysis, enabled us to perform an unbiased identification of the molecular profiles, including ORs expressed, of OSNs activated by specific odorants (*Figure 4A*; *Jiang et al., 2015*; *Hu et al., 2020*).

While the origins of the differences between the scents of mature male and female mice are diverse, we reasoned that mouse urine could be a significant source of sexually dimorphic odor cues. Past literature contrasting intact male mouse urine, castrated male mouse urine, and female mouse urine volatiles has identified many components differing in their presence and abundance (*Novotny et al., 1985*; *Jemiolo et al., 1986*; *Schwende et al., 1986*; *Jemiolo et al., 1991*; *Lin et al., 2005*; *Lin et al., 2007*; *López et al., 2014*). Using pS6-IP-Seq we systematically screened the mouse urine constituents: 2-*sec*-butyl-4,5-dihydrothiazole (SBT), (methylthio)methanethiol (MTMT), β-caryophyllene, 3,4-dehydro-*exo*-brevicomin (DHB), 2,5-dimethylpyrazine (2,5-DMP), (*E*)-β-farnesene, and 2-heptanone (*Figure 4B*).

Differential expression analysis following pS6-IP-Seq across the tested panel of odorants (*Figure 4C–D*, *Figure 4—figure supplement 1A-H*, *Figure 4—figure supplement 2A-D*) led to the identification of the male-specific semiochemical SBT as an agonist for OSNs expressing *Olfr910* or *Olfr912*, and the male-specific semiochemical MTMT as an agonist for OSNs expressing *Olfr1295*. Indeed, exposure to 10 µL of 0.01% (v/v) SBT led to the lowest false discovery rate (FDR) and greatest enrichment of transcripts for *Olfr910* and *Olfr912* from activated OSNs (at an FDR < 0.05), compared to all other ORs, suggesting OSNs expressing *Olfr910* and *Olfr912* to be the most robustly responding OSNs for SBT in vivo (*Olfr910* $\log_2$FC = 2.88 FDR < 1.47E-25, *Olfr912* $\log_2$FC = 3.10 FDR < 2.78E-36; *Figure 4C*). Similarly, exposure to 10 µL of 100 µM MTMT led to the lowest FDR

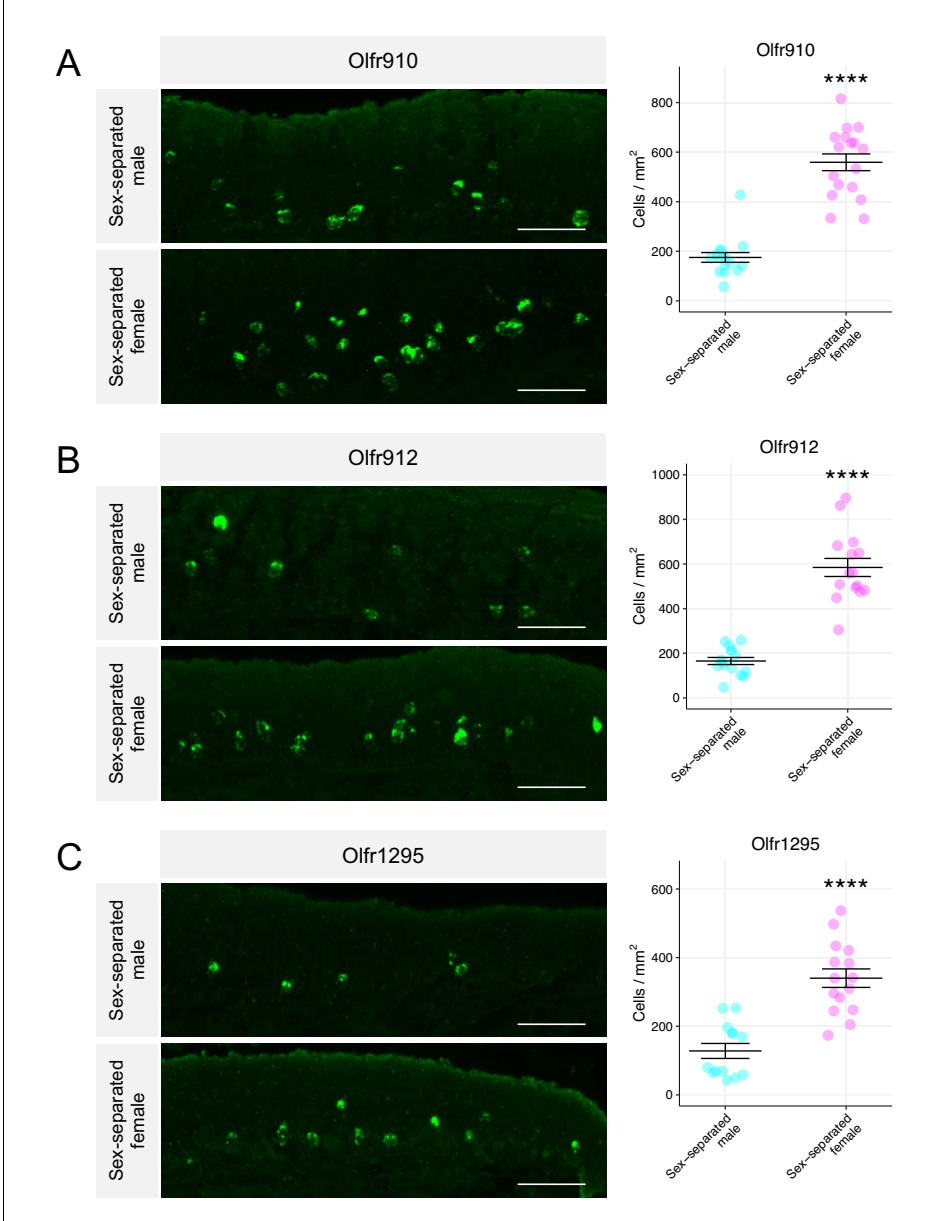

**Figure 2.** Sexually dimorphic expression of ORs is consistent with a change in the number of OSNs expressing those ORs. (**A**) Left: representative *in situ* mRNA hybridization pictures probing for the expression of *Olfr910* in 43-week-old sex-separated male (top) and female (bottom) mice. Scale bars indicate 50 μm. Right: summary data showing mean and SEM of the density of OSNs expressing *Olfr910* in 43-week-old male and female mice. An unpaired two-tailed t-test revealed statistical difference (****p < 0.0001) between males and females. Data are from n = 3 male and n = 3 female mice. (**B**) Representative *in situ* mRNA hybridization pictures probing for the expression of *Olfr912* in 43-week-old sex-separated male (top) and female (bottom) mice. Right: summary data showing mean and SEM of the density of OSNs expressing *Olfr912* in 43-week-old male and female mice. An unpaired two-tailed t-test revealed statistical difference (****p < 0 .0001) between males and females. (**C**) Left: representative *in situ* mRNA hybridization pictures probing for the expression of *Olfr1295* in 43-week-old sex-separated male (top) and female (bottom) mice. Right: summary data showing mean and SEM of the density of OSNs expressing *Olfr1295* in 43-week-old male and female mice. An unpaired two-tailed t-test revealed statistical difference(****p < 0 .0001) between males and females.

The online version of this article includes the following source data and figure supplement(s) for figure 2:

**Source data 1.** Summary statistics for *Figure 2*.

**Figure supplement 1.** Architecture of reads uniquely mapping to female-enriched ORs.

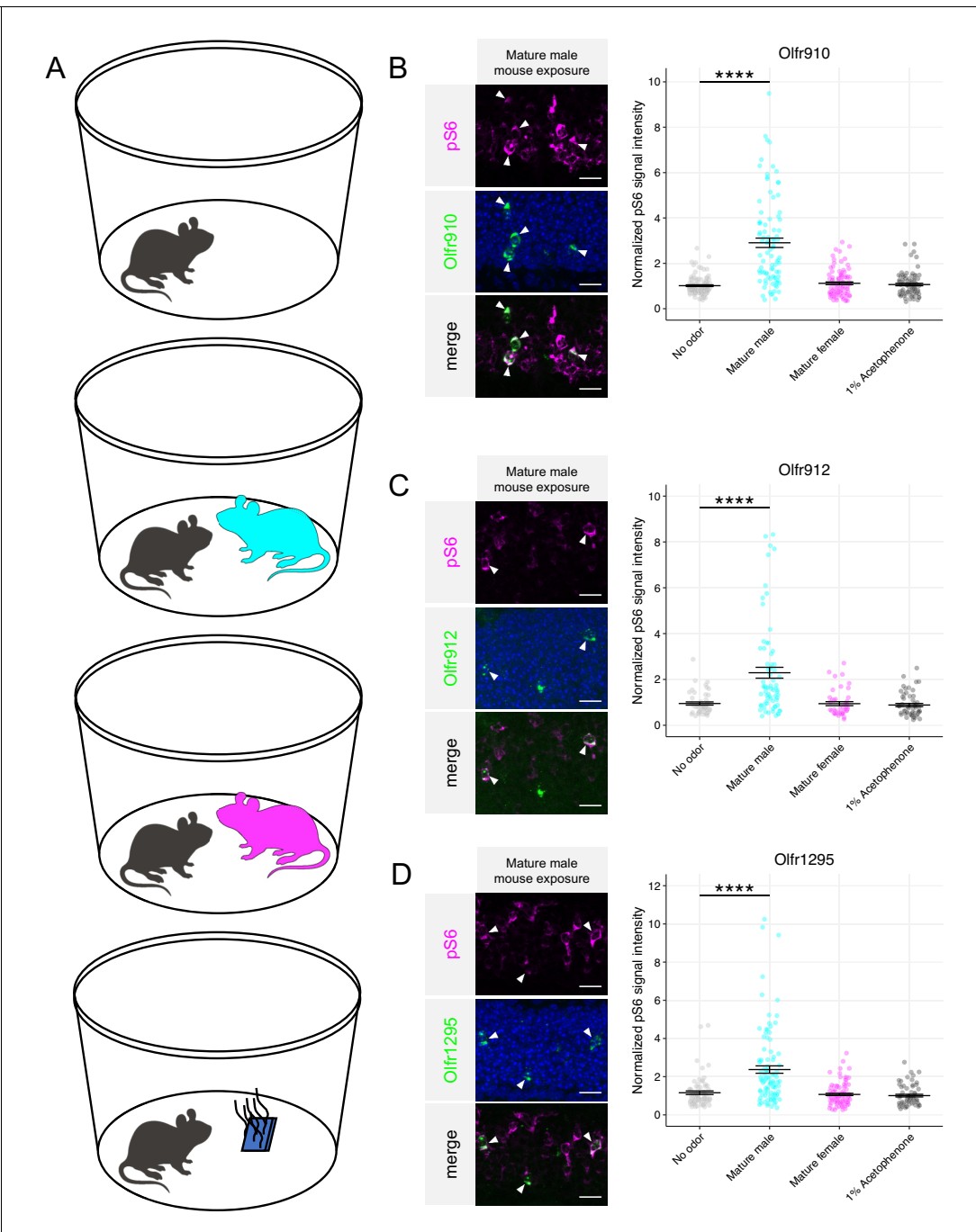

**Figure 3.** Sexually dimorphic ORs are activated by mature male mouse odor. (**A**) Schematic of exposure experiment. A juvenile mouse (black) was exposed to (in descending order) a clean environment, mature male mice, mature female mice, or 1% (v/v) acetophenone for 1 hr in a sealed container. (**B**) Left: representative *in situ* mRNA hybridization and pS6 immunostaining showing co-localization events, as indicated by arrowheads, between OSNs expressing *Olfr910* and pS6 signal induction following exposure of a juvenile mouse to adult male mice. Scale bars indicate 20 μm. Right: summary data showing the mean and SEM of pS6 induction in OSNs expressing *Olfr910* following exposure of a juvenile mouse to multiple stimuli. One-way ANOVA with Dunnett's multiple comparisons test correction reveals that only exposure to mature male mice leads to significant (****p < 0.0001) pS6 induction within OSNs expressing *Olfr910*. Data are from n = 3 juvenile mice. (**C**) Left: representative *in situ* mRNA hybridization and pS6 immunostaining showing co-localization events between OSNs expressing *Olfr912* and pS6 signal induction following exposure of a juvenile mouse to adult male mice. Right: summary data showing the mean and SEM of pS6 induction in OSNs expressing *Olfr912* following exposure of a juvenile mouse to multiple stimuli (****p < 0.0001). (**D**) Left: representative in situ mRNA hybridization and pS6 immunostaining showing co-localization events between OSNs expressing *Olfr1295* and pS6 signal induction following exposure of a juvenile mouse to adult male mice. Right: summary data showing the mean and SEM of pS6 induction in OSNs expressing *Olfr1295* following exposure of a juvenile mouse to multiple stimuli (****p < 0.0001).
*Figure 3 continued on next page*

*Figure 3 continued*

The online version of this article includes the following source data and figure supplement(s) for figure 3:

**Source data 1.** Summary statistics for *Figure 3*.

**Figure supplement 1.** Example *in situ* stainings showing sexually dimorphic ORs are activated by mature male mouse odor.

and greatest enrichment of transcripts for *Olfr1295* from activated OSNs (at an FDR < 0.05), compared to all other ORs, suggesting OSNs expressing *Olfr1295* to be the most robustly responding OSNs for MTMT in vivo (*Olfr1295* $\log_2$FC = 1.80 FDR < 3.72E-5; *Figure 4D*).

To independently validate the pS6-IP-Seq data, we briefly exposed juvenile mice to either an empty odor cassette (control), varying concentrations of SBT or MTMT, or 1% (v/v) acetophenone, and then harvested the MOE for staining. In situ mRNA hybridization probing for the expression of *Olfr910* and *Olfr912* followed by immunostaining for pS6 showed OSNs expressing *Olfr910* and *Olfr912* to display increasingly intense pS6 immunostaining following exposure to SBT in a concentration-dependent manner (*Olfr910*: 0.01% SBT p < 0.0001, 0.1% SBT p < 0.0001, 1% SBT p < 0.0001, one-way ANOVA with Dunnett's multiple comparisons test correction, *Olfr912*: 0.01% SBT p < 0.01, 0.1% SBT p < 0.0001, 1% SBT p < 0.0001, one-way ANOVA with Dunnett's multiple comparisons test correction; *Figure 4E–F*, *Figure 4—figure supplement 3A-B*). Similarly, OSNs expressing *Olfr1295*, identified by in situ mRNA hybridization, displayed increasingly intense pS6 immunostaining following exposure to MTMT, again, in a concentration-dependent manner (100 μM MTMT p < 0.0001, 10 mM MTMT p < 0.0001, one-way ANOVA with Dunnett's multiple comparisons test correction; *Figure 4G*, *Figure 4—figure supplement 3C*). OSNs expressing *Olfr910*, *Olfr912*, or *Olfr1295* also did not display significant pS6 signal intensity differences following exposure to acetophenone compared to control conditions (p > 0.05, one-way ANOVA with Dunnett's multiple comparisons test correction; *Figure 4E–G*), consistent with a lack of enrichment of *Olfr910*, *Olfr912*, and *Olfr1295* by pS6-IP-Seq following exposure to 1% (v/v) acetophenone (*Jiang et al., 2015*; *Hu et al., 2020*).

We then evaluated whether our identified ligand-receptor pairs also exhibited sexually dimorphic responses from OSNs expressing *Olfr910*, *Olfr912*, or *Olfr1295* in mature male and female mice. To do this, we provided a brief exposure of either an empty odor cassette (control), 0.1% (v/v) SBT, 10 mM MTMT, or 1% (v/v) acetophenone to sex-separated, 26-week-old adult male and female mice and then harvested the MOE for staining. Double in situ mRNA hybridization and pS6 immunostaining revealed the following: OSNs expressing *Olfr910* and *Olfr912* showed increasingly intense pS6 signals only following SBT exposure (see summary statistics files; *Figure 5A–B*, *Figure 5—figure supplement 1A-B*), and OSNs expressing *Olfr1295* showed increasingly intense pS6 signals only following MTMT exposure (see summary statistics files; *Figure 5C*, *Figure 5—figure supplement 1C*). We did not observe any sex bias in OSNs expressing *Olfr910*, *Olfr912*, or *Olfr1295* by pS6 signal intensity induction after exposure to either SBT or MTMT, indicating a lack of sexual dimorphism in the response of OSNs to these stimuli (p > 0.05, one-way ANOVA with Tukey's multiple comparisons test correction; *Figure 5A–C*, *Figure 5—figure supplement 1A-C*).

## Long-term cohabitation with the opposite sex is sufficient to attenuate sexual dimorphism in the MOE

Emerging literature has evidenced a role for experience in influencing sensory cell representations within the whole olfactory mucosa (*Xu et al., 2016*; *Ibarra-Soria et al., 2017*; *van der Linden et al., 2018*). Thus, our identification of a subpopulation of over-represented OSNs in sex-separated female mice led us to hypothesize a role for experience in generating this dimorphism.

We hypothesized that regular exposure of a mouse to the opposite sex, by cohabitation, would influence the population dynamics of OSNs exhibiting dimorphic representations. To test this hypothesis, we used mice that were first sex-separated from weaning (~3 weeks age) until 26 weeks of age. These sex-separated mice were then switched to cohabitation with the opposite sex (sex-combined housing) from 26 weeks age to 43 weeks age (*Figure 6A*). At 43 weeks age, whole olfactory mucosa from each male and female mouse was harvested and processed for sequencing and histology.

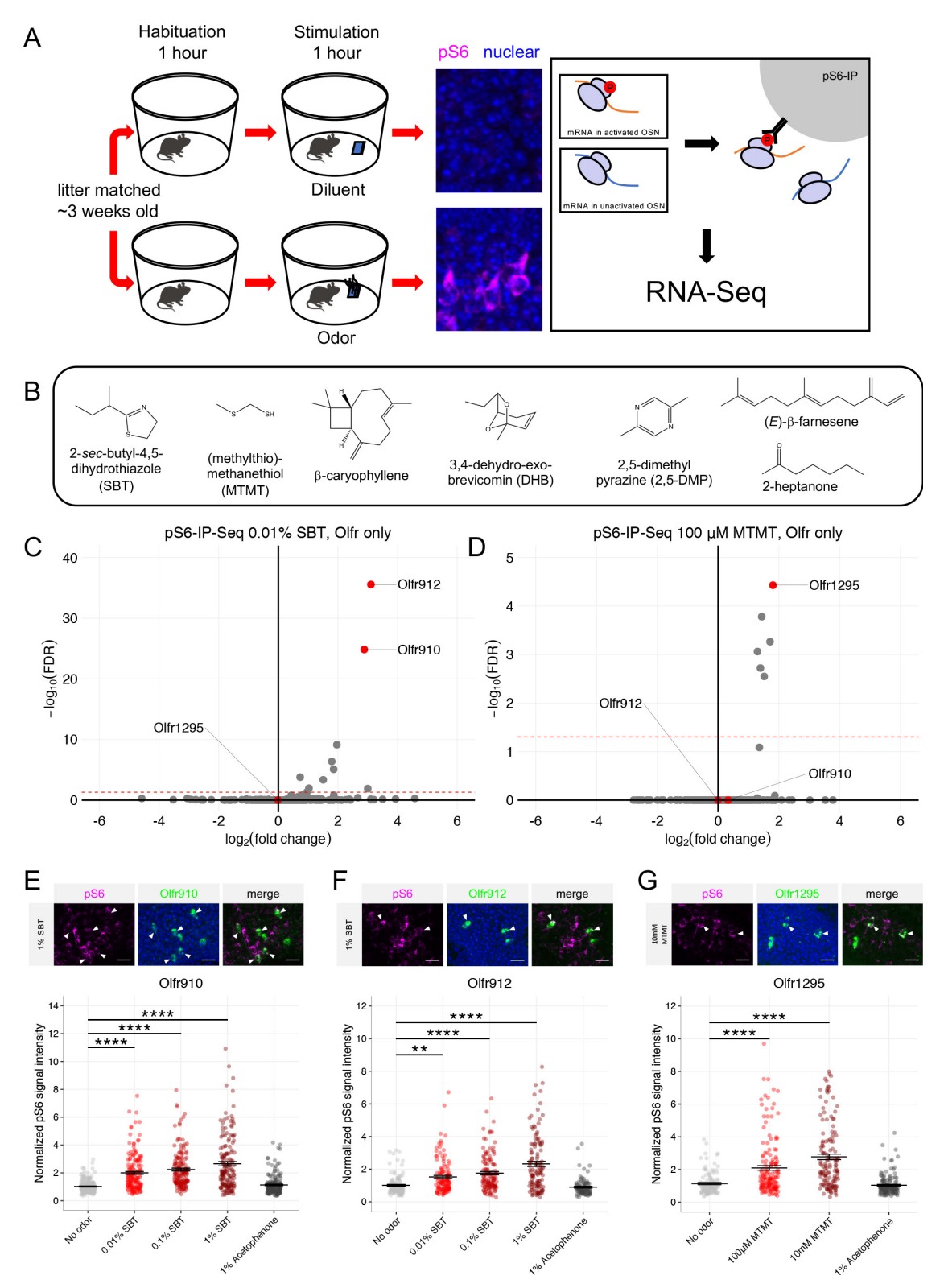

**Figure 4.** Sexually dimorphic ORs are activated by mature male mouse semiochemicals SBT and MTMT. (**A**) Schematic of the pS6-IP-Seq experiment. Litter matched, ~3 week-old (juvenile) mice are used. Mice are habituated to an odor-free environment for 1 hr. One mouse then receives exposure to an odor stimulus, while another receives exposure to the diluent, each for 1 hr. Whole olfactory mucosa is then harvested and immunoprecipitated using an antibody against pS6. (**B**) The panel of sex-specific and sex-enriched volatiles screened using pS6-IP-Seq. (**C**) Volcano plot showing the results

*Figure 4 continued on next page*

*Figure 4 continued*

of pS6-IP-Seq using 0.01% (v/v) SBT diluted in water as stimulus. *Olfr910*, *Olfr912*, and *Olfr1295* are highlighted in red. The red dashed line indicates an FDR = 0.05. Data are from n = 3 control (diluent-exposed) mice and n = 3 experimental (odor-exposed) mice. (D) Volcano plot showing the results of pS6-IP-Seq using 100 μM MTMT dissolved in ethanol as stimulus. (E) Top: representative *in situ* mRNA hybridization and pS6 immunostaining showing co-localization events between OSNs expressing *Olfr910* and pS6 signal induction following exposure of a juvenile mouse to 1% (v/v) SBT diluted in water. Scale bars indicate 20 μm. Bottom: summary data showing the mean and SEM of pS6 induction in OSNs expressing *Olfr910* following exposure of a juvenile mouse to increasing concentrations of SBT and 1% (v/v) acetophenone. One-way ANOVA with Dunnett's multiple comparisons test correction reveals only exposure to 0.01% (v/v) SBT, 0.1% (v/v) SBT, and 1% (v/v) SBT leads to significant pS6 induction within OSNs expressing *Olfr910* (****p < 0.0001). Data are from n = 3 juvenile mice. (F) Top: representative *in situ* mRNA hybridization and pS6 immunostaining showing co-localization events between OSNs expressing *Olfr912* and pS6 signal induction following exposure of a juvenile mouse to 1% (v/v) SBT diluted in water. Bottom: summary data showing the mean and SEM of pS6 induction in OSNs expressing *Olfr912* following exposure of a juvenile mouse to increasing concentrations of SBT and 1% (v/v) acetophenone (**p < 0.01, ****p < 0.0001). (G) Top: representative *in situ* mRNA hybridization and pS6 immunostaining showing co-localization events between OSNs expressing *Olfr1295* and pS6 signal induction following exposure of a juvenile mouse to 10 mM MTMT diluted in ethanol. Bottom: summary data showing the mean and SEM of pS6 induction in OSNs expressing *Olfr1295* following exposure of a juvenile mouse to increasing concentrations of MTMT and 1% (v/v) acetophenone (****p < 0.0001).

The online version of this article includes the following source data and figure supplement(s) for figure 4:

**Source data 1.** Summary statistics for *Figure 4*.
**Figure supplement 1.** Sexually dimorphic ORs are not activated by sex-specific or sex-enriched odorants that are not SBT or MTMT.
**Figure supplement 2.** Cognate ORs for other sex-specific and sex-enriched volatiles are not sexually dimorphic.
**Figure supplement 3.** Example *in situ* stainings showing sexually dimorphic ORs are activated by SBT and MTMT.

Differential expression analysis of ORs from the whole olfactory mucosa from sex-combined male and female mice revealed a severe attenuation of the dimorphic expression of *Olfr910*, *Olfr912*, and *Olfr1295* (at 43 weeks old: *Olfr910* log$_2$FC = 0.80 FDR > 0.65, *Olfr912* log$_2$FC = 0.59 FDR > 0.76,

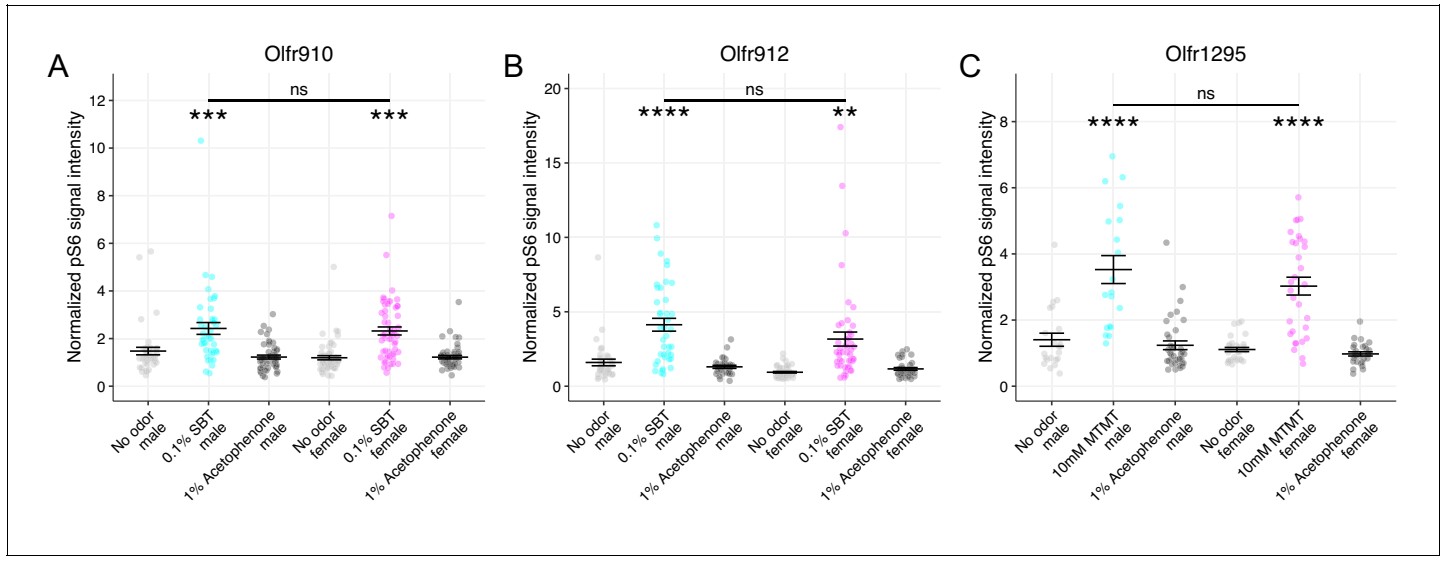

**Figure 5.** OSN responses to semiochemicals are not sexually dimorphic between mature male and female mice. (A) Comparison of responses of OSNs from 26-week-old male and female mice to various stimuli. One-way ANOVA with Tukey's multiple comparisons test correction reveals only exposure to 0.1% (v/v) SBT leads to significant pS6 induction in OSNs expressing *Olfr910* (***p < 0.001) with no significant differences between males and females (ns; p > 0.05). Data are from n = 3 male and n = 3 female mice. (B) One-way ANOVA with Tukey's multiple comparisons test correction reveals only exposure to 0.1% (v/v) SBT leads to significant pS6 induction in OSNs expressing *Olfr912* (**p < 0.01, ****p < 0.0001) with no significant differences between males and females (ns; p > 0.05). (C) One-way ANOVA with Tukey's multiple comparisons test correction reveals only exposure to 10 mM MTMT leads to significant pS6 induction in OSNs expressing *Olfr1295* (****p < 0.0001) with no significant differences between males and females (ns; p > 0.05).

The online version of this article includes the following source data and figure supplement(s) for figure 5:

**Source data 1.** Summary statistics for *Figure 5*.
**Figure supplement 1.** Example *in situ* stainings showing responses to semiochemicals are not sexually dimorphic between mature male and female mice.

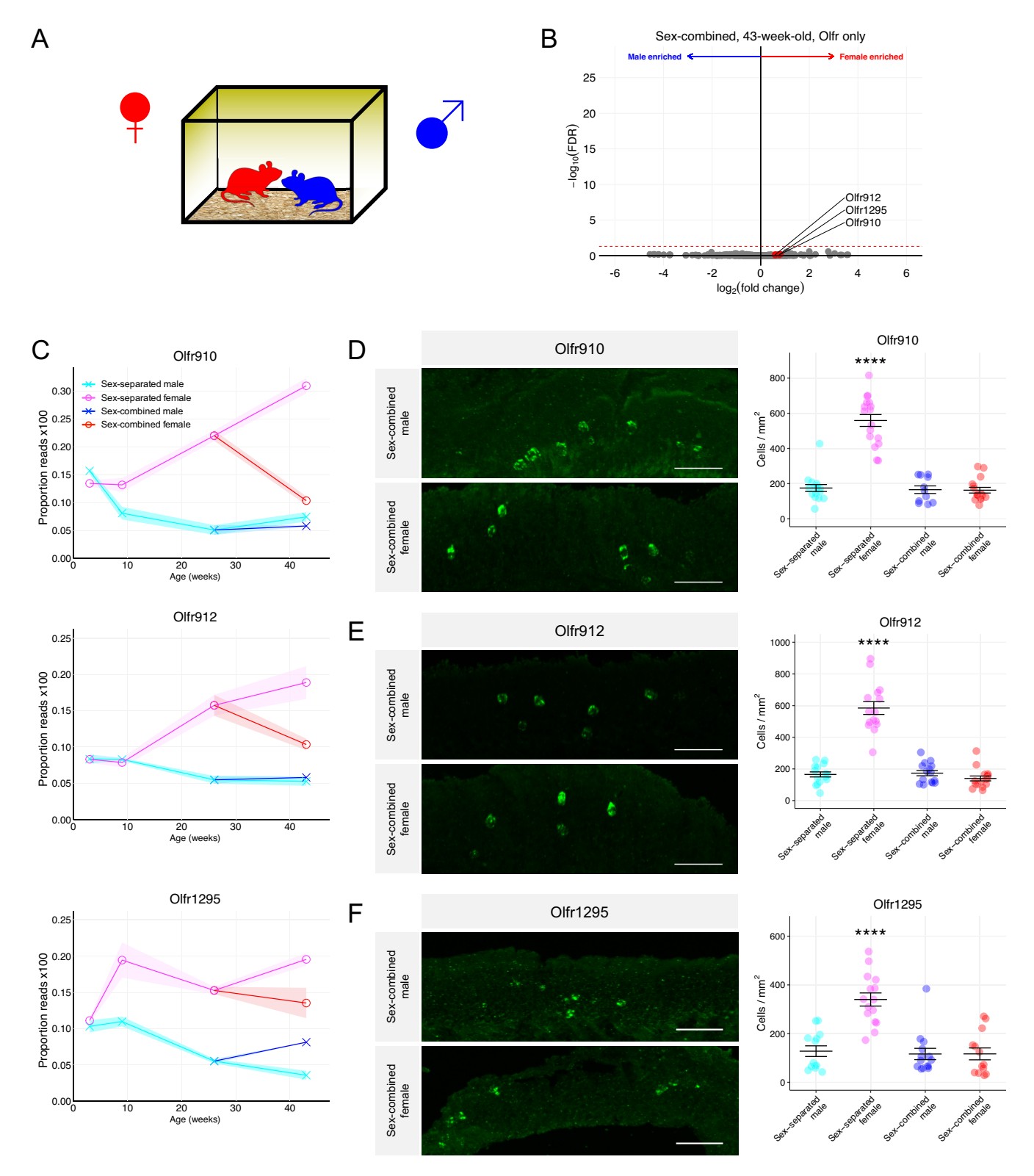

**Figure 6.** Sex-combined housing leads to the attenuation of the dimorphic OR representations. (**A**) Schematic of the housing setup. For sex-combined housing, one male mouse was co-housed with one female mouse. (**B**) Volcano plot comparing expression of *Olfrs* between 43-week-old sex-combined male and female mice. *Olfr910*, *Olfr912*, and *Olfr1295* are highlighted in red. The red dashed line indicates an FDR = 0.05. Data are from n = 3 male and n = 3 female mice. (**C**) Longitudinal plotting of the mean and SEM of proportions of reads aligned to *Olfr910*, *Olfr912*, and *Olfr1295* in sex-

*Figure 6 continued on next page*

Figure 6 continued

separated and sex-combined male and female mice. (D) Left: representative *in situ* mRNA hybridization pictures probing for the expression of *Olfr910* in 43-week-old sex-combined male (top) and female (bottom) mice. Scale bars indicate 50 μm. Right: summary data showing the mean and SEM of the density of OSNs expressing *Olfr910* in 43-week-old male and female mice. One-way ANOVA with Tukey's multiple comparisons test correction reveals only sex-separated female mice to differ in the density of OSNs expressing *Olfr910* (****$p < 0.0001$). Data are from n = 3 male and n = 3 female mice from each housing condition. (E) Left: representative *in situ* mRNA hybridization pictures probing for the expression of *Olfr912* in 43-week-old sex-combined male (top) and female (bottom) mice. Right: summary data showing the mean and SEM of the density of OSNs expressing *Olfr912* in 43-week-old male and female mice. One-way ANOVA with Tukey's multiple comparisons test correction reveals only sex-separated female mice to differ in the density of OSNs expressing *Olfr912* (****$p < 0.0001$). (F) Left: representative *in situ* mRNA hybridization pictures probing for the expression of *Olfr1295* in 43-week-old sex-combined male (top) and female (bottom) mice. Right: summary data showing the mean and SEM of the density of OSNs expressing *Olfr1295* in 43-week-old male and female mice. One-way ANOVA with Tukey's multiple comparisons test correction reveals only sex-separated female mice to differ in the density of OSNs expressing *Olfr1295* (****$p < 0.0001$).

The online version of this article includes the following source data for figure 6:

**Source data 1.** Summary statistics for *Figure 6*.

*Olfr1295* log$_2$FC = 0.76 FDR > 0.70; *Figure 6B*). After 17 weeks of sex-combined housing, the proportional expression of each of these receptors changed much more profoundly in female mice than male mice, again, in a receptor-specific fashion. Normalized expression of *Olfr910* and *Olfr912* appeared to be more similar between sex-separated males, sex-combined males, and sex-combined females while being distinct and less than sex-separated females. On the other hand, the normalized expression of *Olfr1295* appeared to be greatest in sex-separated females, decreasing in sex-combined females, sex-combined males, and lowest in sex-separated males (*Figure 6C*). In situ mRNA hybridization to assess the proportional abundance of OSNs expressing these receptors revealed, again, that sex-separated female mice were distinct from sex-combined female, sex-combined male, and sex-separated male mice. Sex-separated female mice had an over-representation of OSNs expressing *Olfr910*, *Olfr912*, and *Olfr1295* (all $p < 0.0001$, one-way ANOVA with Tukey's multiple comparisons test correction; *Figure 6D–F*), while mice from other conditions were all comparable ($p > 0.05$, one-way ANOVA with Tukey's multiple comparisons test correction; *Figure 6D–F*). These findings demonstrate cohabitation with the opposite sex, potentially via olfactory experience, is sufficient to attenuate the over-representation of the subpopulation of sexually dimorphic OSNs in sex-separated female mice.

## Identification of male mouse over-represented OSNs and cognate ligands

In addition to identifying populations of OSNs over-represented in female mice, our RNA-Seq data also identified over-expression of *Olfr1437* and *Olfr235* in sex-separated male mice. Like female mouse over-represented ORs, *Olfr1437* and *Olfr235* failed to exhibit robust sexually dimorphic expression when mice were housed in a sex-combined fashion (*Figure 7A–C*). The results of in situ mRNA hybridization demonstrated that the proportion of OSNs expressing *Olfr1437* to be greater in 43-week-old male mice than in 43-week-old female mice ($p < 0.0001$, unpaired two-tailed t-test; *Figure 7—figure supplement 1A*). Similar results for *Olfr235* have been shown in *van der Linden et al., 2018*.

In our search for ligands activating sexually dimorphic ORs, we identified the macrocyclic musk odorant cyclopentadecanone (CPD; *Figure 7D*) as an agonist for OSNs expressing *Olfr1437* and *Olfr235* (*Figure 7E*). Importantly, past literature (*McClintock et al., 2014*) has also shown the macrocyclic musk odorant muscone to activate OSNs expressing *Olfr1437* and *Olfr235*. These results suggest musk or musk-related odorants to activate sexually dimorphic, male-enriched populations of OSNs.

## Enrichment of activity-associated genes among sexually dimorphic populations of OSNs

Given the large repertoire of ORs in mice, we found it remarkable that such a small number of ORs exhibited sexually dimorphic expression and representation. We therefore hypothesized OSNs expressing these ORs to harbor a unique gene expression program compared to OSNs expressing

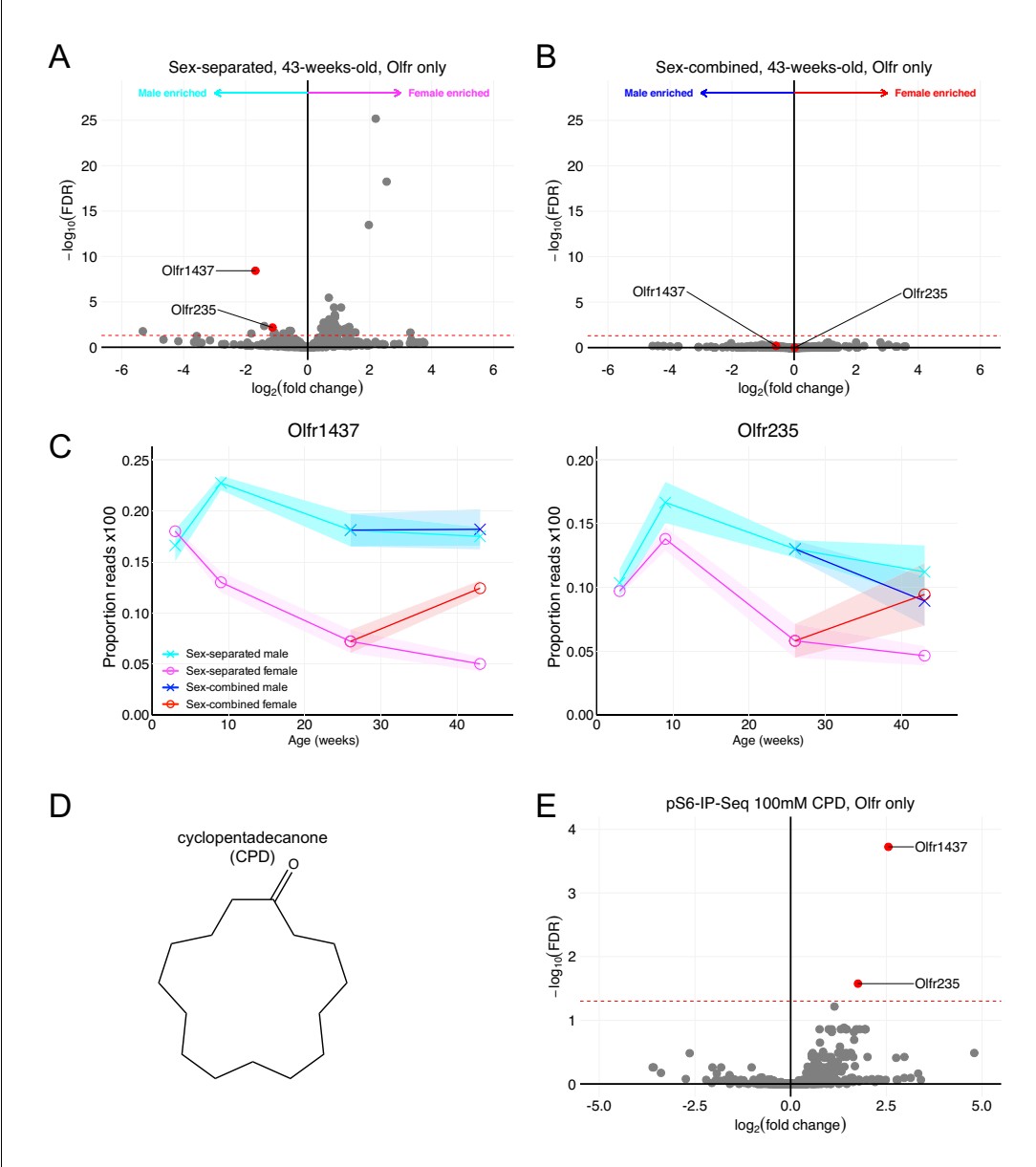

**Figure 7.** *Olfr1437* and *Olfr235* are male-enriched and respond to the macrocyclic musk molecule CPD. (A) Volcano plot comparing expression of *Olfrs* between 43-week-old sex-separated male and female mice. *Olfr1437* and *Olfr235* and are highlighted in red. The red dashed line indicates an FDR = 0.05. Data are from n = 3 male and n = 3 female mice. (B) Volcano plot comparing expression of *Olfrs* between 43-week-old sex-combined male and female mice. (C) Longitudinal plotting of the mean and SEM of proportions of reads aligned to *Olfr1437* and *Olfr235* in sex-separated and sex-combined male and female mice. (D) Structure of macrocyclic musk odorant cyclopentadecanone (CPD). (E) Volcano plot showing the results of pS6-IP-Seq using 100 mM CPD diluted in ethanol as stimulus. *Olfr1437* and *Olfr235* are highlighted in red and enriched. The red dashed line indicates an FDR = 0.05. Data are from n = 3 control (diluent-exposed) mice and n = 3 experimental (odor-exposed) mice.

The online version of this article includes the following source data and figure supplement(s) for figure 7:

**Figure supplement 1.** Male over-expression is also consistent with a difference in the number of cells expressing *Olfr1437* between male and female mice.

**Figure supplement 1—source data 1.** Summary statistics for *Figure 7—figure supplement 1.*

other ORs in a sex-specific manner. To test this hypothesis, we leveraged a recently published single-cell RNA-Seq dataset (*Brann et al., 2020*).

UMAP representations of mature OSN gene expressions, independent of chemosensory receptors, revealed sensory cells expressing *Olfr910*, *Olfr912*, and *Olfr1295* to segregate toward a single

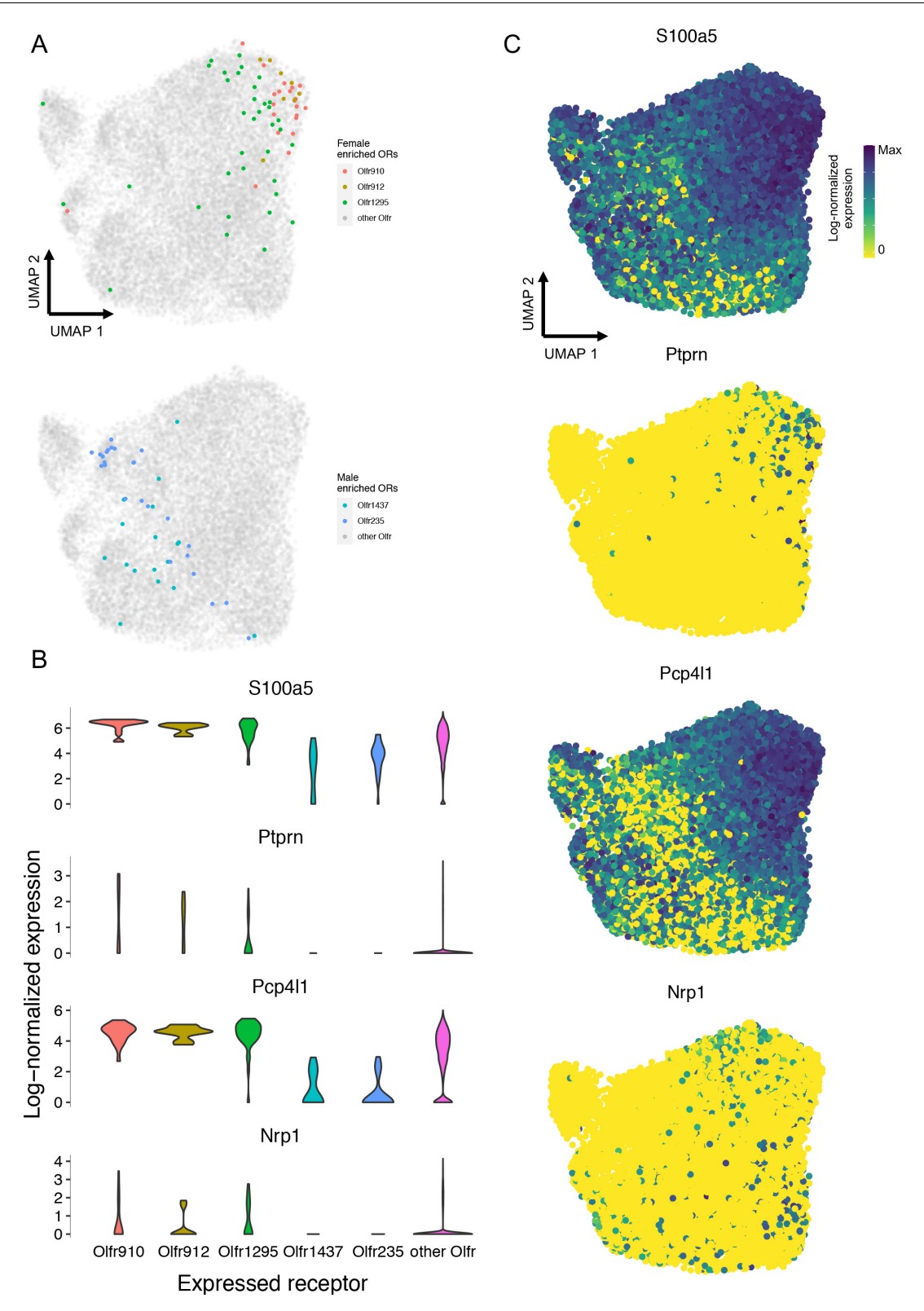

**Figure 8.** Single-cell RNA-Seq analysis reveals an enrichment of activity-associated gene expression in male OSNs expressing *Olfr910*, *Olfr912*, and *Olfr1295*. (A) Chemosensory receptor-independent UMAP embedding of 17,666 singly sequenced mature OSNs. OSNs expressing *Olfr910*, *Olfr912*, *Olfr1295*, *Olfr1437*, and *Olfr235* are highlighted. (B) Gene expression analysis identified activity-associated genes *S100a5*, *Ptprn*, *Pcp4l1*, and *Nrp1* to be

*Figure 8 continued on next page*

*Figure 8 continued*

enriched amongst OSNs expressing *Olfr910, Olfr912,* and *Olfr1295.* (**C**) UMAP representations of *S100a5, Ptprn, Pcp4l1,* and *Nrp1* show a tendency toward higher expression where OSNs expressing *Olfr910, Olfr912,* and *Olfr1295* are localized.

The online version of this article includes the following source data for figure 8:

**Source data 1.** Summary statistics for *Figure 8*.

extreme region while remaining far and distinct from sensory cells expressing *Olfr1437* or *Olfr235* (*Figure 8A*). Expression analysis revealed an enrichment of *S100a5, Ptprn, Pcp4l1,* and *Nrp1* amongst OSNs expressing *Olfr910, Olfr912,* and *Olfr1295.* Importantly, enrichment of these genes had been shown to occur in the literature as a function of neural activity (*Figure 8B–C*; *Imai et al., 2006*; *Nakashima et al., 2013*; *Wang et al., 2017*). Since the singly sequenced mature OSNs from *Brann et al., 2020* were collected from mature male mice, who self-expose to MTMT and SBT, these results are consistent with the idea that odor experience significantly influences gene expression in OSNs, via an activity-dependent process relying on ligand-receptor interactions, to drive dimorphic representations of *Olfr910, Olfr912,* and *Olfr1295*.

## Generating robust sexual dimorphisms in the MOE is *Bax*-dependent

Our observation of specific experiences to influence OSN population dynamics led us to hypothesize a link between OSN activity and neuronal lifespan (*Watt et al., 2004*; *Santoro and Dulac, 2012*). Based on past work showing an association between the elongation of neuronal lifespan and Bax (*Merry and Korsmeyer, 1997*), we hypothesized a role for Bax in influencing sexually dimorphic OR representation by influencing OSN lifespan via an activity-dependent process.

To test the hypothesis of OSN activity influencing OSN lifespan in a Bax-dependent manner, we used *Bax*$^{-/-}$ mice. Compared to wild-type mice, differential expression analysis of RNA sequenced whole olfactory mucosa tissues from sex-separated 26-week-old *Bax*$^{-/-}$ male and female mice revealed an overall lack of sexual dimorphism in the expression of *Olfr910, Olfr912, Olfr1295, Olfr1437,* and *Olfr235* (*Figure 9A*, *Figure 9—figure supplement 1A*). Additionally, in situ mRNA hybridization probing for the proportional abundance of OSNs expressing *Olfr910, Olfr912,* and *Olfr1295,* demonstrated no significant differences between sex-separated *Bax*$^{-/-}$ male and female mice by 43 weeks age ($p > 0.05$, unpaired two-tailed t-test; *Figure 9B–D*). While differential expression analysis comparing *Bax*$^{-/-}$ mice to wild-type mice revealed the expression of many ORs to change (*Figure 9—figure supplement 2A-B*), our results are consistent with the idea that the mechanisms underlying the generation of sexual dimorphism in the MOE are *Bax*-dependent and therefore likely driven by activity-dependent modulations in OSN lifespan.

## Discussion

Using a series of independent but complimentary approaches, we have identified a subpopulation of OSNs, defined by receptor expression, to exhibit sexual dimorphism and experience-dependent plasticity. We have identified female mice, in the absence of a male mouse, to exhibit an over-representation of OSNs expressing *Olfr910, Olfr912,* and *Olfr1295.* Similarly, we have identified male mice, in the absence of a female mouse, to exhibit an over-representation of OSNs expressing *Olfr1437* and *Olfr235.* Long-term cohabitation of mice belonging to opposite sexes led to an overall attenuation of the dimorphic over-representations of these ORs in both sexes. To confirm an OSN activity-dependent component of this phenomenon, we demonstrated female-enriched OSNs to not only be activated by the natural scent of mature male mice, but also to be exquisitely and robustly responsive to the previously identified male-specific semiochemicals SBT and MTMT. In our screening for ligands, we were also able to identify the macrocyclic musk odorant CPD to function as an agonist for male-enriched OSNs. Singly sequenced OSNs expressing *Olfr910, Olfr912,* and *Olfr1295* collected from mature male mice revealed an enrichment of activity-related genes to suggest an axis of activity to influence gene expression in OSNs that may underlie the generation of the robust sexually dimorphic representations of these receptors. Finally, the observation that sex-separated *Bax*$^{-/-}$ mice fail to generate robust sexual dimorphisms in the MOE suggest a role of cell death in generating dimorphic representations of ORs in sex-separated male and female mice.

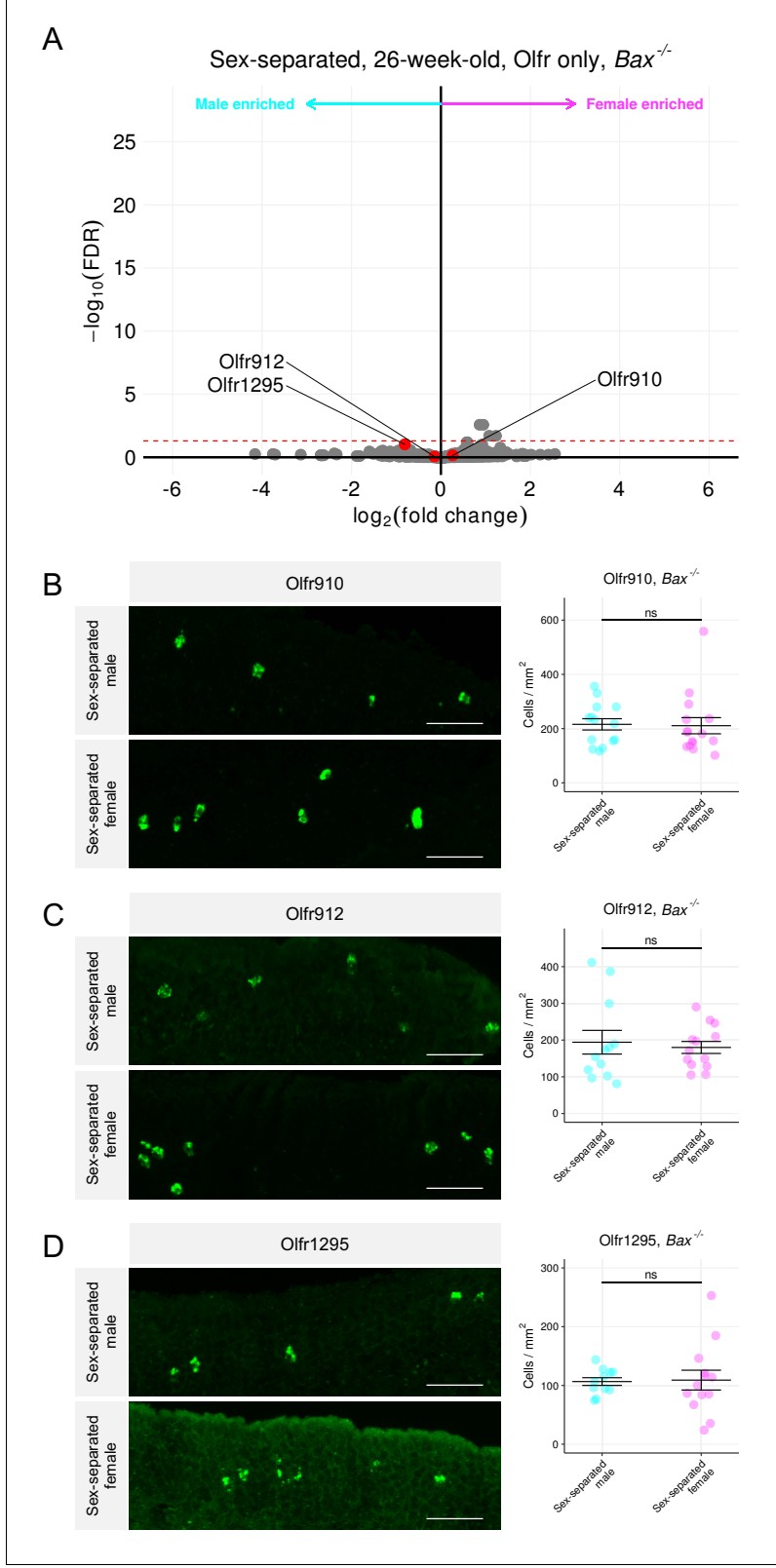

**Figure 9.** Sex-separated *Bax*<sup>-/-</sup> mice fail to generate sexually dimorphic representations of *Olfr910, Olfr912,* and *Olfr1295*. (**A**) Volcano plot comparing expression of *Olfrs* between 26-week-old *Bax*<sup>-/-</sup> sex separated male and female mice. *Olfr910, Olfr912,* and *Olfr1295* are highlighted in red. The red dashed line indicates an FDR = 0.05. Data are from n = 3 male and n = 3 female mice. (**B**) Left: representative *in situ* mRNA

*Figure 9 continued on next page*

*Figure 9 continued*

hybridization pictures probing for the expression of *Olfr910* in *Bax*$^{-/-}$ 43-week-old sex-separated male (top) and female (bottom) mice. Scale bars indicate 50 μm. Right: summary data showing mean and SEM of the density of OSNs expressing *Olfr910* in *Bax*$^{-/-}$ 43 week-old sex-separated male and female mice. An unpaired two-tailed t-test reveals no statistical difference (ns; $p > 0.05$) between males and females. Data are from n = 3 male and n = 3 female mice. (C) Left: representative *in situ* mRNA hybridization pictures probing for the expression of *Olfr912* in *Bax*$^{-/-}$ 43-week-old sex-separated male (top) and female (bottom) mice. Right: summary data showing mean and SEM of the density of OSNs expressing *Olfr912* in *Bax*$^{-/-}$ 43-week-old male and female mice. An unpaired two-tailed t-test reveals no statistical difference between males and females (ns; $p > 0.05$). (D) Left: representative *in situ* mRNA hybridization pictures probing for the expression of *Olfr1295* in *Bax*$^{-/-}$ 43-week-old sex-separated male (top) and female (bottom) mice. Right: summary data showing mean and SEM of the density of OSNs expressing *Olfr1295* in *Bax*$^{-/-}$ 43-week-old male and female mice. An unpaired two-tailed t-test reveals no statistical difference between males and females (ns; $p > 0.05$).

The online version of this article includes the following source data and figure supplement(s) for figure 9:

**Source data 1.** Summary statistics for *Figure 9*.
**Figure supplement 1.** Sex-separated *Bax*$^{-/-}$ mice fail to generate sexually dimorphic representations of *Olfr1437* and *Olfr235*. .
**Figure supplement 2.** *Bax*$^{-/-}$ mice exhibit significant changes in their OR repertoire.

## Olfactory experience as a mechanism to influence OSN population dynamics

Given the capacity of the MOE to regenerate throughout the life of an animal (*Fletcher et al., 2011*; *Brann and Firestein, 2014*), it has been suggested that activity-mediated mechanisms may "individualize" the olfactory system by influencing OSN population dynamics (*Ibarra-Soria et al., 2017*). While we acknowledge cohabitation of the opposite sexes can induce a number of changes in the nervous system of mice (compared to sex-separation) (*Li et al., 2017*; *Remedios et al., 2017*; *Vinograd et al., 2017*), we propose that changes in OSN population dynamics following cohabitation for OSNs expressing *Olfr910*, *Olfr912*, and *Olfr1295*, are in part mediated by olfactory experience with SBT and MTMT. The lengthy timeframes necessary to generate the differences that we observe are consistent with the hypothesis of modulation of OSN lifespan by activity. Nonetheless, we cannot rule out contributions of experience on OSN neurogenesis rates or OR gene choice (*van der Linden et al., 2020*). Questions regarding the individual contributions, or non-contributions, of OSN development and lifespan to generate this dimorphism remain open.

Additionally, the form of plasticity we observe here appears to employ a distinct time course from reports of learning and fear conditioning influencing the glomerular responses (*Kass et al., 2013*; *Abraham et al., 2014*) or proportional abundance of OSNs expressing ORs responsive to the fear-conditioned odor (*Jones et al., 2008*; *Dias and Ressler, 2014*; *Morrison et al., 2015*). In the case of fear conditioning to acetophenone, the number of OSNs expressing M71 (Olfr151) appears to profoundly upregulate within just 3 weeks. In contrast, even by 9 weeks (~6 weeks post-weaning), *Olfr910*, *Olfr912*, and *Olfr1295* are not significantly differentially expressed (*Olfr910* log$_2$FC = 0.54 FDR > 0.24, *Olfr912* log$_2$FC = −0.26 FDR > 0.64, *Olfr1295* log$_2$FC = 0.65 FDR > 0.12; *Figure 1C*). Altogether, these observations lead us to speculate the existence of a multitude of distinct mechanisms, operating at non-identical time scales, to influence OSN population dynamics. Future work to identify and demonstrate these mechanisms is necessary to deepen our understandings of these phenomena and experience-dependent plasticity within the MOE.

## Macrocyclic musk responsiveness in sexually dimorphic male-enriched ORs

Our identification of the macrocyclic musk odorant CPD as an agonist for OSNs expressing the male-enriched ORs *Olfr1437* and *Olfr235* is both intriguing, and consistent, with the previous findings that showed activation of OSNs expressing these receptors by the macrocyclic musk odorant muscone (*McClintock et al., 2014*; *Sato-Akuhara et al., 2016*). Given the sex-specificity of macrocyclic musk odorants in nature, including in the *Moschus* (musk deer) and *Ondatra* (muskrat) species, it has been theorized that this family of odorants may be involved in chemical communication, potentially encoding information about the sex and maturity of an individual (*Li et al., 2016*; *Lee et al.,*

*2019*). Although we were unable to find any literature of sex-specific macrocyclic musk volatile secretion by *Mus musculus*, it is tempting to speculate on the existence of a macrocyclic musk molecule, perhaps generated by female mice, that doubles as a semiochemical.

## A decrease in male-specific semiochemical responsive OSNs in females following sex-combined housing

The finding of over-represented OSN subpopulations robustly responsive to male-specific semiochemicals to decrease in proportional abundance in sex-separated females following sex-combined housing, while consistent with the literature (*Xu et al., 2016*; *Ibarra-Soria et al., 2017*; *van der Linden et al., 2018*), is unexpected. The results suggest that once females receive exposure to male-specific semiochemicals, their detectability for semiochemicals slowly decreases over time, reflecting a potential homeostatic "gain control" mechanism for salient cue detection at the level of primary sensory neurons.

Our further finding that sex-separated $Bax^{-/-}$ mice do not exhibit robust sexually dimorphic representations of ORs point toward a potential role of activity-dependent changes in neural lifespan. Although loss of *Bax* causes changes in the expression of many ORs when compared with wild-type controls (*Figure 9—figure supplement 2A-B*), we speculate the lack of robust sexual dimorphism in sex-separated $Bax^{-/-}$ mice to be a result of the lack of *Bax*-regulated activity-dependent alteration of sensory neuron lifespan. Since $Bax^{-/-}$ mice also show an expanded olfactory epithelial thickness (*Robinson et al., 2003*), the lack of the robust sexually dimorphic representation of ORs may also be a consequence of the expansion of OSNs expressing other ORs responding to self-generated or environmental cues. Future work aimed at understanding the role of OR activity mediated regulation of OSN lifespan and representation will provide mechanistic insights.

## Ventral localization of sexually dimorphic populations OSNs

Despite the early discoveries of ORs being expressed in highly stereotyped locations, the functional consequences of these zonal expression distributions have remained unclear (*Ressler et al., 1993*; *Vassar et al., 1993*). It is interesting to note, all of the identified sexually dimorphic populations of OSNs (*Olfr910*, *Olfr912*, *Olfr1295*, *Olfr1437*, and *Olfr235*) were localized to the ventral region of the MOE (*Tan and Xie, 2018*; *Hu et al., 2020*). These results are consistent with past work showing a robust response in the ventral region of the olfactory bulb to male mouse urine (*Lin et al., 2005*; *Xu et al., 2005*). Based on past findings showing a direct projection from a subset of ventrally localized olfactory bulb mitral cells to the amygdala (*Kang et al., 2009*; *Inokuchi et al., 2017*), it is tempting to speculate that a subset, or even all, of these sexually dimorphic populations of OSNs innervate these previously identified olfactory bulb mitral cells that project to the amygdala.

## The influence of OR activity on OSN gene expression

In our analysis of singly sequenced mature OSNs expressing *Olfr910*, *Olfr912*, and *Olfr1295* harvested from mature male mice we observed a clear enrichment in the expression of numerous activity-associated genes (see summary statistics files; *Figure 8B-C*). Importantly, the expression of many of these same genes was also previously shown to be changed as a function of OSN activity using surgical strategies like unilateral naris occlusion (*Wang et al., 2017*), and transgenic strategies like knock-in expression of constitutively active $G_s$ (*Imai et al., 2006*), and knock-in expression of wild-type and activity-deficient mutant non-OR G-protein coupled receptors (*Nakashima et al., 2013*). Furthermore, these same activity-associated enriched genes were not enriched in singly sequenced mature OSNs expressing *Olfr1437* and *Olfr235*, consistent with the idea that male mice do not generate agonists for these ORs (see summary statistics files; *Figure 8A-C*).

By using an UMAP embedding to examine chemosensory receptor-independent gene expressions of all singly sequenced mature OSNs, we did not observe any obvious and discrete cluster formations. While OSNs expressing *Olfr910*, *Olfr912*, and *Olfr1295* segregated to a single extreme region, they were not removed from the larger embedding itself, and also overlapped with OSNs expressing other ORs. Since UMAP representations emphasize local distances while preserving global structure, these results lead us to speculate that the mechanism by which OR activity influences OSN gene expression to potentially drive the generation of the sexually dimorphic representations of *Olfr910*, *Olfr912*, and *Olfr1295* is not specific to OSNs expressing these ORs. The overlap

between OSNs expressing *Olfr910*, *Olfr912*, and *Olfr1295* with OSNs expressing other ORs in the UMAP embedding also indicates mechanisms like the basal level of OR activity may be implicated in OR-dependent OSN representation. Such a finding would be consistent with the idea of a vectorizable axis of OR activity influencing OSN gene expression and representation.

Similarly, if female mice naturally produce ligands for OSNs expressing *Olfr1437* and *Olfr235*, and the mechanism of the dimorphic representation of *Olfr1437* and *Olfr235* is driven by activity like we speculate for *Olfr910*, *Olfr912*, and *Olfr1295*, then we would expect OSNs expressing *Olfr1437* and *Olfr235* to segregate to a single extreme region when harvested from mature female mice and compared to many other mature OSNs harvested from mature female mice. Future work performing single-cell sequencing on mature OSNs harvested from both male and female mice housed under various conditions will likely provide insights into the specificity of why such a small subset of ORs exhibit experience-dependent sexually dimorphic representations.

## Sexual dimorphism in OSNs responsive to semiochemicals

Plasticity within the OSN population can be postulated to enable adaptation of an individual's olfactory system for the sensitive detection of salient odors, which may vary from one environment to another. While sex-specific chemical cues have been implicated in instinctual behaviors and physiological responses (*Novotny et al., 1985*; *Jemiolo et al., 1986*; *Sam et al., 2001*; *Chamero et al., 2007*; *Haga et al., 2010*; *Dewan et al., 2013*; *Ferrero et al., 2013*; *Li et al., 2013*; *Haga-Yamanaka et al., 2014*; *Fu et al., 2015*; *Hattori et al., 2017*), the degree to which animals are exposed to these chemical cues in nature may vary substantially among individuals.

A report by *van der Linden et al., 2018* also identified sexually dimorphic expression of a subset of ORs using a combination of sequencing and histology-based approaches. Our data agree in the following manner: identification of the sexually dimorphic expression of *Olfr910*, *Olfr912*, *Olfr1295*, *Olfr1437*, and *Olfr235* in mice housed in a sex-separated manner; demonstration of activation of OSNs expressing *Olfr910*, *Olfr912*, and *Olfr1295* following exposure to mature male mice; a general lack of sexual dimorphism in OR expression in mice housed in a sex-combined manner. Together, our findings of experience to influence OSN population dynamics suggest a role of experience in adjusting an animal's sensitivity to salient chemical cue detection.

Our identification of the semiochemicals SBT and MTMT as robust agonists for *Olfr910*, *Olfr912*, and *Olfr1295* posit a number of intriguing speculations. Remarkably, other ORs activated by SBT and MTMT did not exhibit sexual dimorphism. Furthermore, when we tested other sex-specific and sex-enriched odorants, we did not observe activation of OSNs expressing *Olfr910*, *Olfr912*, or *Olfr1295* (*Figure 4—figure supplement 1*), nor did cognate receptors for these other odors exhibit sexually dimorphic expression (*Figure 4—figure supplement 2*). These results altogether lead us to hypothesize a specialized role for *Olfr910*, *Olfr912*, and *Olfr1295* in conveying a salient signal about SBT and MTMT from the olfactory periphery to the central nervous system.

The identification of a subpopulation of OSNs to be plastic and robustly responsive to male-specific semiochemicals also raise speculations about the flexibility of an individual's behavioral responses to semiochemicals. That is, while behavioral and physiological responses to semiochemicals have traditionally been viewed as genetically predetermined and 'hardwired,' there may exist a significant context and experience-dependent flexibility. For example, it has been previously shown that group housed sex-separated female mice exhibit a general suppression and irregularity in estrous cycling. Upon exposure to a mature male mouse, these unisexually grouped female mice often rapidly and synchronously enter into estrus (Whitten effect) (*Whitten, 1956*; *Whitten, 1957*; *Whitten, 1959*; *Whitten et al., 1968*; *Gangrade and Dominic, 1984*). Past implications of SBT also inducing the Whitten effect (*Jemiolo et al., 1986*) along with our finding of *Olfr910* and *Olfr912* to be robustly responsive to SBT and over-represented in sex-separated female mice lead us to speculate that the over-representation of these ORs serves to enhance SBT detection for mediation of the Whitten effect. Past implications of MTMT influencing female mouse attractive behaviors (*Lin et al., 2005*) along with our finding of *Olfr1295* to be robustly responsive to MTMT and over-represented in sex-separated female mice lead us to speculate over-representation of this OR to serve to enhance MTMT detection for mediating attractive responses. Testing these, as well as other possibilities, to link semiochemicals to behavioral and physiological outputs, at the level of molecules, cells, and circuits, remain outstanding.

# Materials and methods

**Key resources table**

| Reagent type (species) or resource | Designation | Source or reference | Identifiers | Additional information |
|---|---|---|---|---|
| Strain, strain background (*Mus musculus* C57BL/6J) | C57BL/6J | Jackson Labs | 000664 | |
| Strain, strain background (*Mus musculus* C57BL/6J) | *Bax*$^{-/-}$ | Jackson Labs | 002994 | |
| Commercial assay, kit | TRIzol | Life Technologies | 15596026 | |
| Commercial assay, kit | QUBIT HS RNA Assay Kit | ThermoFisher | Q32855 | |
| Commercial assay, kit | RNase-free DNaseI | Roche | 04 716 728 001 | |
| Commercial assay, kit | RNeasy Mini Kit | Qiagen | 74104 | |
| Commercial assay, kit | SMART-Seq v4 Ultra Low Input RNA Kit | Takara | 634898 | |
| Commercial assay, kit | Nexterra XT DNA Library Preparation Kit | Illumina | 15032354 | |
| Sequence-based reagent | Olfr910 UTR F | This paper | | 5'-AAACGCGTGTGAAAATT GTGACAGATCCA-3' |
| Sequence-based reagent | Olfr910 UTR R | This paper | | 5'-AAGCGGCCGCCATTTAC AAGAAGGGAATCAG-3' |
| Sequence-based reagent | Olfr912 UTR F | This paper | | 5'-AAACGCGTACTTTGTTC TGATTCAGTTGTT-3' |
| Sequence-based reagent | Olfr912 UTR R | This paper | | 5'-AAGCGGCCGCGTCCAC AGAGCAATACAACA-3' |
| Sequence-based reagent | Olfr1295 UTR F | This paper | | 5'-AAACGCGTACTCCTCT CCTAAATCCAAC-3' |
| Sequence-based reagent | Olfr1295 UTR R | This paper | | 5'-AAGCGGCCGCGGCAG CACCACTGATCAA-3' |
| Commercial assay, kit | Phusion DNA Polymerase | NEB | F530S | |
| Commercial assay, kit | pCI vector | Promega | E1731 | |
| Commercial assay, kit | DNA polymerase | Qiagen | 203203 | |
| Commercial assay, kit | MinElute Kit | Qiagen | 28004 | |
| Commercial assay, kit | T3 RNA polymerase | Promega | P2083 | |
| Commercial assay, kit | DIG RNA labeling mix | Roche | 11277073910 | |
| Commercial assay, kit | Micro Bio-Spin P-30 Gel Columns | Bio-Rad | 732–6223 | |
| Commercial assay, kit | Tissue-Tek O.C.T. Compound | Sakura Finetek | 4583 | |
| Commercial assay, kit | Superfrost Plus Slides | Fisherbrand | 1255015 | |
| Commercial assay, kit | Parafilm | Sigma | P7793 | |

*Continued on next page*

*Continued*

| Reagent type (species) or resource | Designation | Source or reference | Identifiers | Additional information |
|---|---|---|---|---|
| Commercial assay, kit | Blocking reagent | Roche | 11096176001 | |
| Commercial assay, kit | Maleic acid | Sigma | M0375 | |
| Antibody | Anti-DIG antibody (sheep polyclonal) | Roche | 11207733910 | 1:1000 |
| Commercial assay, kit | TSA-Fluorescein | PerkinElmer | NEL741B001KT | 1:400 |
| Antibody | pS6 Antibody for IF (rabbit polyclonal) | ThermoFisher | 44–923G | 1:300 |
| Antibody | Anti-rabbit IgG Cy3 Antibody (donkey polyclonal) | Jackson Immuno | 711-165-152 | 1:200 |
| Other | Bisbenzimide | Sigma | H 33258 | 1:100,000 |
| Other | Paper bucket | International Paper | DFM85 | |
| Other | Blotting pad | VWR | 28298–014 | |
| Other | Odor cassette | Sakura Finetek | 0006772–01 | |
| Commercial assay, kit | LoBind Tube | Eppendorf | 22431021 | |
| Commercial assay, kit | NP40 Substitute | Sigma | 11332473001 | |
| Commercial assay, kit | DHPC | Avanti Polar Lipids | 850306P | |
| Antibody | pS6 Antibody for IP (rabbit monoclonal) | Cell Signaling | D68F8 | 6 µL |
| Commercial assay, kit | Dynabeads Protein A | Invitrogen | 10002D | |
| Commercial assay, kit | RNeasy Micro Kit | Qiagen | 74004 | |
| Other | β-Caryophyllene | Sigma | W225207 | |
| Other | 2,5-DMP | Sigma | 175420 | |
| Other | 2-Heptanone | Sigma | W254401 | |
| Other | (*E*)-β-Farnesene | Bedoukian | P3500-90 | |
| Other | CPD | Sigma | C111201 | |
| Other | MTMT | *Lin et al., 2005* | | Synthesized |
| Other | SBT | *Meijer et al., 1973*; *Abrunhosa et al., 2001*; *Tashiro and Mori, 1999* | | Synthesized |
| Other | DHB | *Wiesler et al., 1984* | | Synthesized |

## Animal husbandry

Wild-type C57BL/6J (Jackson Labs 000664) and *Bax*<sup>-/-</sup> (Jackson Labs 002994) were bought and maintained at institutional facilities. Procedures of animal handling and tissue harvesting were approved by the Institutional Animal Care and Use Committee of Duke University. Animals were killed within 7 days of the ages reported in this study. Sex-separated male and female mice were socially housed with two to five animals per cage. Sex-combined cages contained one male and one female. All sex-combined cages produced litters. Pups were aged to P21-P28 before being weaned or used for independent experiments.

Three male and three female biological replicates were used in each condition to sequence wild-type and *Bax*<sup>-/-</sup> whole olfactory mucosa tissues (MOE + other tissues in the nose). Three male and three female wild-type and *Bax*<sup>-/-</sup> mice were used in each condition to examine MOE in situ.

## Preparation of olfactory tissues for RNA-Seq

Whole olfactory mucosa was rapidly collected in 5 mL tubes and flash-frozen in liquid nitrogen from mice killed by $CO_2$ asphyxiation and decapitation. Tissues were kept frozen at $-80$ °C until time of RNA extraction. To extract RNA, 1 mL of TRIzol (Life Technologies 15596026) was added to frozen tissue followed by homogenization until no large pieces were readily identifiable. Homogenized tissue was transferred to new 1.5 mL tubes and centrifuged at max speed for 10 min. Supernatant was then transferred to new 1.5 mL tubes containing 0.2 mL chloroform and vortexed for 3 min. Samples were again centrifuged at max speed for 15 min and the aqueous phase was transferred to new tubes containing 0.5 mL of isopropanol. Samples were incubated at room temperature for 5 min and then centrifuged at max speed for 10 min. Supernatant was decanted and the visible pellet was washed 150 µL of 75% ethanol, centrifuged, and washed again with 180 µL of 75% ethanol. After centrifugation, ethanol wash was pipetted away and RNA pellets were allowed to air-dry with tube lids kept open for 10 min. Pellets were then dissolved in RNase-free water by heating to 55 °C for 10 min. RNA concentration was quantified using a QUBIT HS RNA Assay Kit (ThermoFisher Q32855).

Eighty-eight µL of sample was subjected to RNase-free DNaseI treatment by the addition of 10 µL of 10X Buffer and 2 µL of RNase-free DNaseI (Roche 04 716 728 001) for 30 min at 37 °C. Following DNaseI treatment, samples were subjected to a modified RNeasy mini protocol for RNA cleanup (Qiagen 74104). A total of 350 µL of buffer RLT was added to the 100 µL sample, mixed, and centrifuged. Then, 250 µL ethanol was added, mixed, and immediately transferred to a mini-column. Sample loaded columns were centrifuged for 30 s. Five hundred µL of ethanol diluted buffer RPE was then used to wash the column twice, and sample was eluted in new 1.5 mL tubes with 100 µL of RNase-free water. Presence of RNA was confirmed by the QUBIT HS RNA Assay Kit.

Amplified cDNA from RNA was prepared using a SMART-Seq v4 Ultra Low Input RNA Kit (Takara 634898) protocol as per the manufacturer's guidelines. In the case of whole olfactory mucosa sequencing, two rounds of cDNA amplification were used with 1000 ng of input RNA. DNA libraries were prepared using a half-sized Nexterra XT DNA Library Preparation Kit (Illumina 15032354) protocol as per the manufacturer's guidelines. Libraries were sequenced on either HiSeq 2000/2500 (50 base pair single read mode) or NextSeq 500 (75 base pair single read mode) with 6–12 pooled indexed libraries per lane. Reads were aligned against a modified GRCm38.p6 (M25) reference, in which we deleted ENSMUSG00000116179 (Olfr290), using STAR (*Dobin et al., 2013*) with –outFilterMultimapNmax 10. Reads mapping to Olfr290 were inferred from ENSMUSG00000070459, with the rationale that this gene model included ENSMUSG00000116179 plus untranslated regions. Gene-level read quantification was done using RSEM (*Li and Dewey, 2011*). Differential expression analysis was performed against all genes using EdgeR (*Robinson et al., 2010*). Gene nomenclature was retrieved from BioMart (*Smedley et al., 2009*). Intact *Olfr* genes were filtered, and p-values were then re-corrected by FDR. Receptor-specific read proportions were calculated by comparing reads mapped to the specific *Olfr*s compared to those mapped to other *Olfr*s. Raw and processed datasets generated as part of this study are available from NCBI GEO at accession GSE160272.

## Cloning OR coding and untranslated region sequences

To identify unique 3′ UTR regions of *Olfr910*, *Olfr912*, and *Olfr1295*, all whole olfactory mucosa RNA-Seq fastq files were first concatenated in a sex-specific manner. Reads were then uniquely aligned using STAR using –outFilterMultimapNmax 1. BAM files were sorted and indexed for read mapping visualization using Samtools (*1000 Genome Project Data Processing Subgroup et al., 2009*). Regions at the 3′ end of the coding sequence, that were occurring within defined transcript variants of each of the ORs were then individually subjected to NCBI BLAST until a unique region of at least 500 bases was identified. To each of the forward primers a 5′-AA-3′ and 5′-ACGCGT-3′(MluI restriction site) was appended. To each of the reverse primers a 5′-AA-3′ and 5′-GCGGCCGC-3′ (NotI restriction site) was appended. The final primer sequences were as follows: *Olfr910* forward: 5′-AAACGCGTGTGAAAATTGTGACAGATCCA-3′, *Olfr910* reverse: 5′-AAGCGGCCGCCATTTA-CAAGAAGGGAATCAG-3′, *Olfr912* forward: 5′-AAACGCGTACTTTGTTCTGATTCAGTTGTT-3′, *Olfr912* reverse: 5′-AAGCGGCCGCGTCCACAGAGCAATACAACA-3′, *Olfr1295* forward: 5′-AAACGCGTACTCCTCTCCTAAATCCAAC-3′, *Olfr1295* reverse: 5′-AAGCGGCCGCGGCAGCAC-CACTGATCAA-3′. An identical methodology was used to clone the coding sequence of *Olfr1437*.

Regions of interest were amplified from genomic DNA using Phusion (NEB F530S) as per the manufacturer's guidelines. Amplified fragments were cloned into pCI expression vectors (Promega E1731) containing the first 20 residues of human rhodopsin (Rho-pCI) and were verified by sequencing.

## Generation of anti-sense RNA probes

To generate anti-sense digoxigenin (DIG)-RNA probes, ORFs were amplified (Qiagen 203203) from Rho-pCI vectors and purified via a MinElute Kit (Qiagen 28004) using manufacturer protocols with an added T3 polymerase promoter sequence at the 3' end. Anti-sense RNA was then in vitro transcribed using a T3 RNA polymerase (Promega P2083) and a DIG RNA labeling mix (Roche 11277073910) using manufacturer protocols. RNA probes were then alkaline hydrolyzed (80 mM NaHCO$_3$, 120 mM Na$_2$CO$_3$) for 60 °C for 15 min and purified using a microcolumn (Bio-Rad 732–6223). Probe integrity was assessed by agarose gel and kept at −80 °C when not in use.

## Preparation of olfactory tissues for staining and in situ mRNA hybridization

For all adult mouse experiments, since *Olfr910*, *Olfr912*, *Olfr1295*, *Olfr1437*, and *Olfr235* are all expressed ventrally (*Tan and Xie, 2018*), sensory epithelium attached to the septum was rapidly micro-dissected and frozen in embedding medium (Sakura Finetek 4583) from mice killed by CO$_2$ asphyxiation and decapitation. For juvenile mice, full snouts were prepared. Eighteen to 22 µm fresh frozen coronal sections were cut using a cryostat (Leica CM1850) onto microscope slides (Fisherbrand Superfrost Plus 1255015) and kept at −80 °C until use.

For in situ RNA probe hybridization, slides were brought to room temperature, dried and rapidly fixed in 4% paraformaldehyde in 1x PBS (pH 7–8) for 15 min. Slides were then washed twice 1x PBS and submerged into a triethanolamine solution consisting of 700 mL dH$_2$O with 8.2 mL triethanolamine. 1.75 mL of acetic anhydride was then added dropwise over the course of 7 min with constant and slow stirring for a total of 10 min, all at room temperature. Slides were then washed with 1x PBS and blocked with prehybridization solution consisting of 50% formamide (Invitrogen 15515-026), 5x SSC, 250 µg/mL Baker's yeast RNA (Sigma R6750), 100 µg/mL herring sperm DNA (Sigma D7290), 1 mM DTT, and 300 U/mL heparin (Sigma H3393) for 1 hr at 58 °C in a humidified hybridization oven. RNA probe concentrations were then individually optimized by dilution in prehybridization buffer and pipetted directly onto slides and covered with laboratory film (Parafilm 54956) for overnight hybridization at 58 °C. Slides were then rinsed the next day in 72 °C heated 5x SSC twice, washed twice in 72 °C heated 0.2x SSC for 30 min per wash, and again finally washed in 1x PBS for a minimum of 5 min at room temperature. Slides were then blocked with 0.5% nucleic acid blocking reagent (Roche 11096176001) dissolved in a 1x PBS containing maleic acid (Sigma M0375) for 30 min. Blocking solution was then replaced with 1:1000 horseradish peroxidase (HRP)-conjugated anti-DIG antibody (Roche 11207733910) solution diluted in the blocking medium for 45 min. Slides were then washed in 1x PBS three times, 10 min per wash, and coated with 0.1% BSA in 1x PBS. Hybridization signals were detected using tyramide signal amplification (TSA) with TSA-Fluorescein (PerkinElmer NEL741B001KT) as the fluorophore diluted 1:400 in 1x PBS containing 0.003% H$_2$O$_2$ via incubation for 10 min in darkness, all at room temperature.

For pS6 immunostaining, slides were blocked in 5% skim milk dissolved in 1x PBS containing 0.1% Triton X-100 at room temperature for 1 hr. Blocking solution was then replaced with 1:300 anti-pS6 antibody (ThermoFisher 44–923G) dissolved in blocking solution and incubated overnight at 4 °C. Anti-pS6 antibody was detected using a 1:200 Cy3-conjugated secondary (Jackson Immuno 711-165-152) diluted in 5% skim milk dissolved in 1x PBS by incubation for 45 min in darkness. Cell nuclei were detected using a 1/10,000 dilution of a 1% Bisbenzimide (Sigma Bisbenzimide H 33258) solution by incubation for 5 min at room temperature. All slides were rinsed in dH$_2$O, cover slipped, and allowed to dry before examination under a microscope.

## Slide microscopy and image quantification

Z-stacked images with 2 µm intervals between each slice were obtained at 200 $\times$ magnification using the Zeiss Axiocam MRm and upright inverted fluorescent microscope with ApoTome functionality. The filter sets used were as follows: Zeiss filter set #38 for Fluorescein, #43 for Cy3, and #49 for

Bisbenzimide. For OSN density quantification, areas were visually drawn in ImageJ around the OSN layer of the sensory epithelia in each image. In situ mRNA hybridization positive cells were counted to report a density of specific OSNs per unit area. Data are reported as the density of OSNs expressing specific receptors per imaged 20-µm section as individual data points. For pS6 signal intensity quantification, Cy3 signals (pS6 intensity) within Fluorescein-positive cells (OR expression) were merged as a maximum intensity projection in ImageJ. Average pS6 signal intensities within single cells were then normalized by average pS6 signal intensities across all OSNs in the OSN layer of the sensory epithelia within the same image to report a normalized single neuron pS6 signal intensity as an individual data point.

## Odor exposures

All juvenile mice used in this study were ~3 weeks old. Mice were habituated for 1 hr in a clean and covered single-use paper container (International Paper DFM85) in a fume hood. For odor exposure, mice were then transferred to a new paper container containing either the mature male mice (~8 weeks old), mature female mice (~8 weeks old), or diluted odorant for another hour. All odorant exposures in this study consisted of 10 µL of stimulus spotted onto a cut blotting pad (VWR 28298–014) placed inside an odor cassette (Sakura Finetek 0006772–01). All odorants, unless otherwise stated, were diluted in water, vortexed, and rapidly spotted. MTMT and CPD were diluted in ethanol. Control experiments for odor exposures consisted of exposing mice to water or ethanol spotted on blotting pads placed in odor cassettes. Odor exposure experiments on juvenile mice employed both male and female mice. Odor exposure experiments on mature mice used both males and females. Mature females used were at or near estrus determined by vaginal cell cytology (*Byers et al., 2012*).

## Phosphorylated S6 ribosomal capture (pS6-IP)

Mice used for pS6-IP were ~3 weeks old, mixed sex, littermates. Mice were killed by $CO_2$ asphyxiation and cervical dislocation. Olfactory tissue was rapidly dissected in Buffer B (2.5 mM HEPES KOH pH 7.4, 0.63% glucose, 100 µg/mL cycloheximide, 5 mM sodium fluoride, 1 mM sodium orthovanadate, 1 mM sodium pyrophosphate, 1 mM β-glycerophosphate, in Hank's balanced salt solution). Tissue pieces were then minced in 1.35 mL Buffer C (150 mM KCl, 5 mM $MgCl_2$, 10 mM HEPES KOH pH 7.4, 0.100 µM Calyculin A, 2 mM DTT, 100 U/mL RNAsin, 100 µg/mL cycloheximide, protease inhibitor cocktail, 5 mM sodium fluoride, 1 mM sodium orthovanadate, 1 mM sodium pyrophosphate, 1 mM β-glycerophosphate)and subsequently transferred to homogenization tubes for steady homogenization at 250 rpm three times and at 750 rpm nine times at 4 ˚C. Samples were then transferred to a 1.5 mL LoBind tube (Eppendorf 022431021) and clarified at 2000xg for 10 min at 4 ˚C. The low-speed supernatant was transferred to a new tube on ice, and to this solution was added 90 µL of NP40 (Sigma 11332473001) and 90 µL of 1,2-diheptanoyl-sn-glycero-3-phosphocholine (DHPC, Avanti Polar Lipids 850306P, 100 mg/0.69 mL). This solution was mixed and then clarified at a max speed (17,000xg) for 10 min at 4 ˚C. The resulting high-speed supernatant was transferred to a new tube where 20 µL was saved and transferred to a tube containing 350 µL buffer RLT. To the remainder of the sample, 1.3 µL of 100 µg/mL cycloheximide, 27 µL of phosphatase inhibitor cocktail (250 mM sodium fluoride, 50 mM sodium orthovanadate, 50 mM sodium pyrophosphate, 50 mM β-glycerophosphate) and 6 µL of anti-pS6 antibody (Cell Signaling D68F8) was added. The sample was gently rotated for 90 min at 4 ˚C. To prepare beads, 100 µL of beads (Invitrogen 10002D) was washed three times with 900 µL of buffer A (150 mM KCl, 5 mM $MgCl_2$, 10 mM HEPES KOH pH 7.4, 10% NP40, 10% BSA), and once with 500 µL of buffer C. Sample homogenate was added to the beads and incubated with gentle rotation for 60 min at 4 ˚C. Following incubation, beads were washed with four times with 700 µL of buffer D (350 mM KCl, 5 mM $MgCl_2$, 10 mM HEPES KOH pH 7.4, 10% NP40, 2 mM DTT, 100 U/mL RNAsin, 100 µg/mL cycloheximide, 5 mM sodium fluoride, 1 mM sodium orthovanadate, 1 mM sodium pyrophosphate, 1 mM β-glycerophosphate). During the final wash, beads were moved to room temperature, wash buffer was removed, and 350 mL of buffer RLT was added. Beads were incubated in buffer RLT for 5 min at room temperature. Buffer RLT containing immunoprecipitated RNA was then eluted and stored at −80 ˚C until clean up using a kit (Qiagen 74004). cDNA was generated using 11 rounds of amplification with 10 ng RNA input.

## Source of odorants

MTMT was synthesized as previously described in *Lin et al., 2005*. Racemic SBT was synthesized in two steps from 2-aminoethanol and methyl 2-methylbutanedithioate (*Meijer et al., 1973*) in >80% yield (*Abrunhosa et al., 2001*). The final product showed high purity by gas chromatography, $^1$H and $^{13}$C NMR, and mass spectroscopic data. The data were in agreement with reported parameters for this compound (*Tashiro and Mori, 1999*). DHB was synthesized as previously described (*Wiesler et al., 1984*). β-Caryophyllene (Sigma W225207), 2,5-DMP (Sigma 175420), 2-heptanone (Sigma W254401), (*E*)-β-farnesene (Bedoukian P3500-90), and CPD (Sigma C111201) were purchased.

## Single-cell RNA-Seq analysis

Unique molecular identifier (UMI) counts were downloaded from NCBI GEO accession GSE151346. Mature OSNs were subset into a Seurat (v 3.2.2) object (*Butler et al., 2018*). For UMAP dimensionality reduction, mitochondrial genes, and genes beginning with "Olfr" and "Taar" were pre-filtered. Highly variable and predicted olfactory receptor genes "OR5BS1P", "Gm13762", "Gm13723", and "Gm13757" were also pre-filtered. UMI counts were then log-normalized, scaled, and top variable genes identified by the "vst" selection method were examined to confirm no chemosensory receptor or predicted chemosensory receptor genes. The dimensionality of the data was then reduced via principal component analysis. Eighteen dimensions were used as input for two dimensional UMAP visualization.

Mature OSNs were each assigned an intact olfactory receptor defined by the greatest number of raw UMI counts for that cell. To identify genes enriched, cells expressing *Olfr910*, *Olfr912*, and *Olfr1295* were pooled (A), cells expressing *Olfr1437* and *Olfr235* were pooled (B), cells expressing *Olfr910*, *Olfr912*, *Olfr1295*, *Olfr1437*, and *Olfr235* were pooled (C). Pools A, B, and C were then individually differentially expressed to all other mature OSNs. Only genes with an average log fold change > 0 and p-adjusted < 0.05 were considered. Further analysis was then focused on the subset of enriched genes that were then also consistently found in the literature to be implicated in activity-dependent processes. Shown genes were also cross validated by differentially expressing normalized and scaled UMI counts of single receptor expressing OSNs (post-filtering of "Olfr", "Taar", and predicted olfactory receptors) to all mature OSNs to confirm an average log fold change > 0 and p-adjusted value < 0.05 in at least one of the cell types (see summary statistics for Figure 8).

## Acknowledgements

We thank lab members Claire A de March, Ha Na Choe, Conan Juan, Yen Dinh, Tatjanna Abaffy, and Maira Nagai for comments on the manuscript, Mengjue Jessica Ni for expert technical assistance, and Douglas Marchuk and Debra Silver for generous sharing of equipment.

## Additional information

### Competing interests

Hiroaki Matsunami: HM receives royalties from Chemcom. The other authors declare that no competing interests exist.

### Funding

| Funder | Grant reference number | Author |
| --- | --- | --- |
| National Institute on Deafness and Other Communication Disorders | DC014423 | Eric Block<br>Hiroaki Matsunami |
| National Institute on Deafness and Other Communication Disorders | DC016224 | Hiroaki Matsunami |
| National Science Foundation | 1556207 | Hiroaki Matsunami |

The funders had no role in study design, data collection and interpretation, or the decision to submit the work for publication.

## Author contributions
Aashutosh Vihani, Conceptualization, Data curation, Software, Formal analysis, Validation, Investigation, Visualization, Methodology, Writing - original draft, Project administration, Writing - review and editing; Xiaoyang Serene Hu, Data curation; Sivaji Gundala, Sachiko Koyama, Resources; Eric Block, Resources, Funding acquisition; Hiroaki Matsunami, Conceptualization, Resources, Supervision, Funding acquisition, Validation, Investigation, Methodology, Project administration, Writing - review and editing

## Author ORCIDs
Aashutosh Vihani (iD) https://orcid.org/0000-0001-5979-101X
Sachiko Koyama (iD) https://orcid.org/0000-0002-6886-1961
Eric Block (iD) https://orcid.org/0000-0002-4668-1687
Hiroaki Matsunami (iD) https://orcid.org/0000-0002-8850-2608

## Ethics
Animal experimentation: This study was performed in strict accordance with the recommendations in the Guide for the Care and Use of Laboratory Animals of the National Institutes of Health. All of the animals were handled according to approved institutional animal care and use committee (IACUC) protocol A128-19-06 at Duke University.

## Decision letter and Author response
Decision letter https://doi.org/10.7554/eLife.54501.sa1
Author response https://doi.org/10.7554/eLife.54501.sa2

# Additional files
## Supplementary files
• Transparent reporting form

## Data availability
Sequencing data have been deposited in GEO under accession codes GSE160272.

The following datasets were generated:

| Author(s) | Year | Dataset title | Dataset URL | Database and Identifier |
|---|---|---|---|---|
| Vihani A, Matsunami H | 2020 | Semiochemical responsive olfactory sensory neurons are sexually dimorphic and plastic [RNA-Seq of whole olfactory mucosa] | https://www.ncbi.nlm.nih.gov/geo/query/acc.cgi?acc=GSE160270 | NCBI Gene Expression Omnibus, GSE160270 |
| Vihani A, Matsunami H | 2020 | Semiochemical responsive olfactory sensory neurons are sexually dimorphic and plastic [pS6-IP-Seq of whole olfactory mucosa] | https://www.ncbi.nlm.nih.gov/geo/query/acc.cgi?acc=GSE160271 | NCBI Gene Expression Omnibus, GSE160271 |

The following previously published dataset was used:

| Author(s) | Year | Dataset title | Dataset URL | Database and Identifier |
|---|---|---|---|---|
| Brann DH, Tsukahara T, Datta S | 2020 | Non-neuronal expression of SARS-CoV-2 entry genes in the olfactory system suggests mechanisms underlying COVID-19-associated | https://www.ncbi.nlm.nih.gov/geo/query/acc.cgi?acc=GSE151346 | NCBI Gene Expression Omnibus, GSE151346 |

anosmia (scRNA-seq: olfactory mucosa)

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
