## [Decision Letter]

Thank you for submitting your article "Semiochemical responsive olfactory sensory neurons are sexually dimorphic and plastic" for consideration by *eLife*. Your article has been reviewed by three peer reviewers, and the evaluation has been overseen by a Reviewing Editor and Catherine Dulac as the Senior Editor. The following individual involved in review of your submission has agreed to reveal their identity: Stavros Lomvardas (Reviewer #1).

The reviewers have discussed the reviews with one another and the Reviewing Editor has drafted this decision to help you prepare a revised submission.

The authors investigate sexually dimorphic patterns of odorant receptor gene expression in the mouse olfactory epithelium. The authors reveals that three olfactory receptor genes (*Olfr910, 912* and *1295*) have sexually dimorphic expression that depends upon long-term sex separation.

Specifically these ORs are overrepresented in female mice relative to male mice, when female mice are housed together with females over several months. Also they find that co-housing male and female mice reduces the number of female neurons expressing these three receptors to levels similar to the levels of expression in males. Importantly, pS6-IP-RNA seq identifies *Olfr 910* and *912* as odorant receptors responsive to SBT, and *Olfr 1295* as responsive to MTMT, two semiochemicals present in male mouse urine. Finally, they show that the expression of these three receptors is not sexually dimorphic in *Bax* KO mice, suggesting that prolonged exposure to male semiochemicals accelerates the rate of apoptosis in *Olfr910, 912* and *1295*.

All the reviewers agree that the findings are interesting and novel (in most parts). However, they also think that some data need further in-depth analyses and quantification, as well as suggest of possible few additional experiments to better support the findings and conclusion. For example, repeat the ISH with more specific probes over numerous animals with a clear description of how the count and normalize across sections. Characterize the *Olfr 910, 912*, and *1295* subtypes of sensory neurons using in situ hybridization or immuno-fluorescence experiments with known pro- and anti-apoptotic factors. As an alternative, the authors could use their pS6 RNA-seq data or, if available, scRNA-seq data to identify molecular differences of these neuronal subtypes.

For your convenience below are the lightly edited comments of the three reviewers to convey these essential revisions.

Reviewer #1:

I would appreciate a bit more information on the unique characteristics of the neurons that express these three OR genes. Obviously I am not asking for a whole new set of experiments, but if they could perform some in situ hybridization or immuno-fluorescence experiments with known pro- and anti-apoptotic factors that may expressed at different levels in these neuronal subtypes, this would be insightful. Alternatively, if the authors have scRNA-seq data that they could harness towards identifying molecular differences of neurons that express these three receptors of if they could use the pS6 RNA-seq data to identify genes enriched in the semiochemical-responding populations vs populations that respond to other ligands. In this note, I am curious if they could compare the RNA seq libraries of wt and *Bax* KO mice and identify additional OR genes that have altered expression frequency in *Bax* KO mice. Again, I am not asking for a whole new set of experiments, just for a more comprehensive analysis of their existing RNA-seq data.

Finally, I have a technical/conceptual concern. In Figure 2, the authors show a clear sexual dimorphism in neurons expressing *olfr910/912* in 43 weeks, with male epithelia having ~350 neurons/mm2 and female epithelia having ~900 neurons/mm2. Further, when male and female mice are co-housed, the numbers of female *olfr910/912*-expressing neurons goes down to ~350 neurons/mm2. This suggests that prolonged exposure of *olfr910/912*-expressing neurons to their cognate semiochemicals increases their rate of apoptosis, predicting that blocking apoptosis would increase the number of *olfr910/912*-expressing neurons in males. However, this is not the result in the *Bax* KO mice, where both male and female mice have ~350 neurons/mm2. I understand that the role of *Bax* in apoptosis is complex, with described roles as pro- and anti- apoptotic functions in the nervous system. However, neither pro- or anti-apoptotic *Bax* function could explain why both co-housing with males and deleting *Bax* would result in reduction of *olfr910/912*-expressing neurons in females. The same concerns are true for *olfr1295*. Unless there is a trivial explanation that I am missing, I believe that there are complicating variables (such as hormonal fluctuations, indirect effects from co-housing, genomic background issues) that would require more than a year to resolve given that it takes 43 week to obtain the full effect in sexual dimorphic OR expression.

Reviewer #2:

Specific points:

1) The quantification of RNA in situ and pS6 signals from histological sections is insufficient. The authors need to consistently analyze at least 3 mice per condition. Individual odorant receptors are non-homogeneously expressed across the surface of the olfactory epithelium. The authors need to show that the positions of regions of the olfactory epithelium used for quantification are comparable across individuals and conditions. Exposure of one mouse to another mouse for an hour results in extensive olfactory exploration, which likely increases sensory neuron activity across a large fraction of the odorant receptor repertoire. Overall changes in sensory neuron activity should be reported to validate the specificity of *Olfr 910, 912*, and *1295* activation. The authors need to specify what the individual data points in their figures represent.

2) In situ probes against odorant receptors were designed against the odorant receptor coding region, which, as the authors point out, show substantial sequence similarity. The authors state that odorant receptor expression was determined by visual identification of the highest and most intense cells positive for the RNA in situ probe. The authors need to show histological images comprising areas with cells showing these different levels of “positivity”, to document the specificity of their probes. Ideally, more specific in situ probes should be designed against non-homologous 5' or 3' UTRs.

3) The observation that sexual dimorphism of odorant receptor gene expression is attenuated in *Bax*-deficient mice lacks in depth analysis and interpretation. Can the authors exclude that the observed effects are caused by differences in the genetic background of mice? The authors state that neural activity is thought to promote neural survival. *Bax* is a pro-apoptotic protein required for programmed cell death. One simple prediction thus is that exposure of *Olfr 910, 912*, and *1295* to male semiochemicals promotes their survival, and that this overrepresentation of *Olfr 910, 912*, and *1295* is further enhanced in a *Bax*-deficient background. The authors need to provide a plausible explanation for their observations, or better, identify the molecular pathways involved in *Bax*-dependent modulation of odorant receptor gene expression.

Minor comments:

1) Housing females with males results in pregnancy and the birth of offspring. Pregnancy and motherhood are well known to induce robust changes in odor perception and could have effects on the expression of the odorant receptor repertoire. The authors should discuss this possibility.

2) The authors should comment on the identity and experience-dependent regulation of odorant receptors that are overrepresented in male mice.

3) All supplemental figures should be referenced in the main text.

4) The authors discuss their work in the broad context of inter-individual variability. Sexual dimorphism is not an example of inter-individual variability. The authors may want to rephrase the general context of this study.

Reviewer #3:

The mechanisms of activity-dependent regulation of the ORs are not quite clear to me. I recommend the following relatively straight forward experiments and analysis to strengthen the paper.

1) It is not clear how *BAX* enables activity-dependent regulation of the three receptor types. From Figure 7B, it appears there are more ORs in males, but quantification indicates a decrease in cells expressing these receptors when compared with other figures. The authors should clarify this by clearly state the effect of *BAX* knockout on the number of cells. They may also examine the expression levels of *olfr910/912* and *olfr1295* in the RNAseq data to further quantify the change.

2) If *BAX* is expressed broadly, how does it control the activity-dependent survival of only a few receptor types? It is not clear how this specificity is achieved. The authors may address this issue by examining whether *BAX* is expressed at a higher level by neurons expressing these receptors. If not, the authors should discuss the involvement of other potential mechanisms.

---

## [Author Response]

We thank all reviewers for their feedback. We also appreciate the patience of all individuals involved in the review process of this manuscript. Since we exhausted almost all tissues from the original submission, we had to prepare new mice for experiments. It has been extremely challenging to perform all the suggested experiments with our long-term mouse housing scheme and mouse housing scheme manipulations during the Coronavirus pandemic.

Reviewer #1:I would appreciate a bit more information on the unique characteristics of the neurons that express these three OR genes. Obviously I am not asking for a whole new set of experiments, but if they could perform some in situ hybridization or immuno-fluorescence experiments with known pro- and anti-apoptotic factors that may expressed at different levels in these neuronal subtypes, this would be insightful. Alternatively, if the authors have scRNA-seq data that they could harness towards identifying molecular differences of neurons that express these three receptors of if they could use the pS6 RNA-seq data to identify genes enriched in the semiochemical-responding populations vs populations that respond to other ligands. In this note, I am curious if they could compare the RNA seq libraries of wt and Bax KO mice and identify additional OR genes that have altered expression frequency in Bax KO mice. Again, I am not asking for a whole new set of experiments, just for a more comprehensive analysis of their existing RNA-seq data.

Though we were unable to perform single-cell RNA-Seq experiments in our lab, we have leveraged a recently published mouse single-cell RNA-Seq dataset, Brann et al., 2020, in which the authors profiled more than 17,000 individual mature OSNs from mature male mice. Through re-analysis of this dataset, we were able to consistently observe enrichment of numerous genes previously shown to be enriched as a function of activity amongst OSNs expressing *Olfr910*, *Olfr912*, and *Olfr1295*. As such, we have also expanded our Results and Discussion sections.

We also tried to use pS6-IP-Seq data to identify candidate genes, beyond olfactory receptors, differentially expressed in semiochemical responsive OSNs. However, it is difficult to interpret these results, since it is unclear if the differential expression analysis would be a consequence of activity-dependent enrichment following pS6-IP or native to the OSN. Additionally, when we cross-checked these genes with data from single-cell sequencing of mature OSNs, non-chemosensory receptor genes enriched by pS6-IP-Seq are not necessarily robustly expressed by mature OSNs.

To maximize the interpretations of *Bax^-/-^* mice in the context of our presented work, we originally sought to limit our focus in the main figures to *Olfr910*, *Olfr912*, and *Olfr1295*. We however agree with the reviewer that showing differences between wild-type and *Bax^-/-^* mice in terms of general OR expression is important. We now include plots differentially expressing RNA-Seq libraries generated from wild-type and *Bax^-/-^* mice Figure 9—figure supplement 2. It can be seen that hundreds of ORs are indeed differentially expressed between mutant and wild-type mice. The interpretation of these results, in totality, is difficult because it appears the expression alteration of each OR is independent of the others. Nonetheless, as the data clearly show, sex-separated *Bax^-/-^* mice do not exhibit sexually dimorphic expression of *Olfr910*, *Olfr912*, *Olfr1295*, *Olfr1437*, and *Olfr235*. We propose that understanding the role of *Bax*, beyond our focus of experience-dependent sex-specific receptor expression, to be outside the scope of this manuscript. We also now include some further discussions highlighting the complexities of that may arise from *Bax* deficiency in OR expression:

“Our further finding that sex-separated *Bax^-/-^* mice do not exhibit robust sexually dimorphic representations of ORs point towards a potential role of activity-dependent changes in neural lifespan. Though loss of *Bax* causes changes in the expression of many ORs when compared with wild-type controls, (Figure 9—figure supplement 2A-B) we speculate the lack of robust sexual dimorphism in sex-separated *Bax^-/-^* mice to be a result of the lack of *Bax*-regulated activity-dependent alteration of sensory neuron lifespan. Since *Bax^-/-^* mice also show an expanded olfactory epithelial thickness, the lack of the robust sexually dimorphic representation of ORs may also be a consequence of the expansion of OSNs expressing other ORs responding to self-generated or environmental cues. Future work aimed at understanding the role of OR activity mediated regulation of OSN lifespan and representation will provide mechanistic insights.”

Finally, I have a technical/conceptual concern. In Figure 2, the authors show a clear sexual dimorphism in neurons expressing olfr910/912 in 43 weeks, with male epithelia having ~350 neurons/mm2 and female epithelia having ~900 neurons/mm2. Further, when male and female mice are co-housed, the numbers of female olfr910/912-expressing neurons goes down to ~350 neurons/mm2. This suggests that prolonged exposure of olfr910/912-expressing neurons to their cognate semiochemicals increases their rate of apoptosis, predicting that blocking apoptosis would increase the number of olfr910/912-expressing neurons in males. However, this is not the result in the Bax KO mice, where both male and female mice have ~350 neurons/mm2. I understand that the role of Bax in apoptosis is complex, with described roles as pro- and anti- apoptotic functions in the nervous system. However, neither pro- or anti-apoptotic Bax function could explain why both co-housing with males and deleting Bax would result in reduction of olfr910/912-expressing neurons in females. The same concerns are true for olfr1295. Unless there is a trivial explanation that I am missing, I believe that there are complicating variables (such as hormonal fluctuations, indirect effects from co-housing, genomic background issues) that would require more than a year to resolve given that it takes 43 week to obtain the full effect in sexual dimorphic OR expression.

Indeed, we are also perplexed by this observation. Since the expression frequency of hundreds of OR genes are altered in *Bax^-/-^* mice compared to wild-type mice, examination of the proportion reads aligned to each receptor is not easily interpretable. To attempt to mitigate some of these concerns we have now done in situ mRNA hybridization experiments using 3’ UTR sequences specific to each of the ORs. However, in this scenario now, the proportional abundance of *Olfr910*, *Olfr912*, and *Olfr1295* in sex-separated *Bax^-/-^* mice, in both sexes, resemble the wild-type sex-separated male mice, wild-type sex-combined male mice, and wild-type sex-combined female mice. While our data is consistent with a *Bax*-dependency in generating the sexual dimorphism, it is still unclear to us how to interpret this result in its entirety. We speculate that the full-body knockout of *Bax* influences OR expression by multiple mechanisms, as we now have included in the main text. We propose however that resolving these mechanisms are outside the scope of this manuscript. Nonetheless, we believe the results from the *Bax^-/-^* mice the are still very informative of a potential underlying mechanism, and therefore continue to present the results, but we are also very conservative about the interpretations and implications of the *Bax* experiments throughout the manuscript.

Reviewer #2:Specific points:1) The quantification of RNA in situ and pS6 signals from histological sections is insufficient. The authors need to consistently analyze at least 3 mice per condition. Individual odorant receptors are non-homogeneously expressed across the surface of the olfactory epithelium. The authors need to show that the positions of regions of the olfactory epithelium used for quantification are comparable across individuals and conditions. Exposure of one mouse to another mouse for an hour results in extensive olfactory exploration, which likely increases sensory neuron activity across a large fraction of the odorant receptor repertoire. Overall changes in sensory neuron activity should be reported to validate the specificity of Olfr 910, 912, and 1295 activation. The authors need to specify what the individual data points in their figures represent.

We have now repeated the experiments using n=3 mice for all *in situ* experiments. Importantly, the majority of our conclusions are identical, and effect sizes have increased and become more robust.

For all of our data generated (in the original submission and revision), using adult mice, we have consistently focused our analysis on the septum of the MOE to maximize consistently between mice. We have made this clearer in our methods section (lines 522 - 525):

For all of our data generated (in the original submission and revision), using adult mice, we have consistently focused our analysis on the septum of the MOE to maximize consistently between mice. We have made this clearer in our Materials and methods section:

“For all adult mouse experiments, since Olfr910, Olfr912, Olfr1295, Olfr1437, and Olfr235 are all expressed ventrally (Tan and Xie, 2018), sensory epithelium attached to the septum was rapidly micro-dissected and frozen in embedding medium (Tissue-Tek O.C.T. Compound 4583) from mice killed by CO_2_ asphyxiation and decapitation.”

We also did this because the adult MOE is filled with calcified bones which cannot be sectioned. By micro-dissecting the septum we were able to probe for OSNs expressing ORs across ventral zones using in situ mRNA hybridization. We performed some preliminary experiments by first trying to decalcify adult MOE, to allow for sectioning, but in our hands, the decalcification and pre-fixing procedure significantly reduces signals from *in situ* mRNA hybridization. We therefore concluded that it would be unfeasible to do this in conjunction with *in situ* mRNA hybridization targeting 3’ UTR unique OR sequences which are much more lowly expressed than coding sequence segments.

We agree with the reviewer that indeed exposure of a mouse to monomolecular odors or other mice induces a general increase in activity throughout the MOE measured by pS6 signal intensity though pS6 induction in OSNs expressing *Olfr910*, *Olfr912*, and *Olfr1295* are specific to SBT (*Olfr910* and *Olfr912*) and to MTMT (*Olfr1295*). Indeed, our data showing pS6 signal intensities of OSNs expressing *Olfr910*, *Olfr912*, and *Olf1295* are normalized against the pS6 signal intensity in the OSN layer of the MOE in the same image. Our Materials and methods now more clearly explain this:

“Average pS6 signal intensities within single cells were then normalized by average pS6 signal intensities across all OSNs in the OSN layer of the sensory epithelia within the same image to report a normalized single neuron pS6 signal intensity…”

Our Materials and methods also now more clearly describe how all individual data points were generated for the figures in the manuscript.

“In situ mRNA hybridization positive cells were counted to report a density of specific OSNs per unit area. Data are reported as the density of OSNs expressing specific receptors per imaged 20-micron section as individual data points. For pS6 signal intensity quantification, Cy3 signals (pS6 intensity) within Fluorescein positive cells (OR expression) were merged as a maximum intensity projection in ImageJ. Average pS6 signal intensities within single cells were then normalized by average pS6 signal intensities across all OSNs in the OSN layer of the sensory epithelia within the same image to report a normalized single neuron pS6 signal intensity as an individual data point.”

2) In situ probes against odorant receptors were designed against the odorant receptor coding region, which, as the authors point out, show substantial sequence similarity. The authors state that odorant receptor expression was determined by visual identification of the highest and most intense cells positive for the RNA in situ probe. The authors need to show histological images comprising areas with cells showing these different levels of “positivity”, to document the specificity of their probes. Ideally, more specific in situ probes should be designed against non-homologous 5' or 3' UTRs.

We thank the reviewer for the critical suggestion. We have redone all experiments using uniquely designed 3’ UTR probes and n=3 mice throughout the manuscript in an attempt to fully mitigate concerns of sequence similarity/overlap of ORs. Details of the 3’ UTR design are described in both Figure 2—figure supplement 1 and in the Materials and methods section of the paper. Unique probe design was verified not only by NCBI BLAST, but also by cross-validation within our own RNA-Seq data. Importantly, our data now show even more robust differences between control and experimental conditions and are more consistent with quantifications based on RNA-Seq.

3) The observation that sexual dimorphism of odorant receptor gene expression is attenuated in Bax-deficient mice lacks in depth analysis and interpretation. Can the authors exclude that the observed effects are caused by differences in the genetic background of mice? The authors state that neural activity is thought to promote neural survival. Bax is a pro-apoptotic protein required for programmed cell death. One simple prediction thus is that exposure of Olfr 910, 912, and 1295 to male semiochemicals promotes their survival, and that this overrepresentation of Olfr 910, 912, and 1295 is further enhanced in a Bax-deficient background. The authors need to provide a plausible explanation for their observations, or better, identify the molecular pathways involved in Bax-dependent modulation of odorant receptor gene expression.

We cannot experimentally exclude the possibility that the observed effects are caused by the genetic background of the mice. However, the mouse line we used (Jackson Labs 002994) is a congenic strain which has been backcrossed over 20 times to C57BL/6 according to the Jackson Labs website.

We also agree with this reviewer (and reviewer 1 and 3) that the interpretations of *Bax* are complex. Nonetheless, we have expanded our Discussion surrounding the results and implication of these results. We still however limit ourselves to what our data most directly shows: generation of the sexually dimorphic expression of *Olfr910*, *Olfr912*, *Olfr1295*, *Olfr1437*, and *Olfr235* are *Bax*-dependent.

We also agree with the reviewer that identifying the molecular pathway leading to a *Bax*-dependent modulation of odorant receptor expression frequency would be ideal. However, this would be a series of experiments that would likely take years.

Minor comments:1) Housing females with males results in pregnancy and the birth of offspring. Pregnancy and motherhood are well known to induce robust changes in odor perception and could have effects on the expression of the odorant receptor repertoire. The authors should discuss this possibility.

Indeed, there are many possibilities to explain changes in receptor expression frequency. We believe that activity, via experience, is the primary driver and therefore emphasize this in our main text. We are unfamiliar with any literature showing a direct influence of pregnancy and motherhood in influencing OR representation. Furthermore, in van der Linden et al. 2018, which also reported changes in the expression of Olfr910, Olfr912, Olfr1295, Olfr1437, and Olfr235 using a similar paradigm, the authors killed all pups within 1 day after birth. Nonetheless, we acknowledge this possibility and cite an article previously showing plasticity in the olfactory system associated with motherhood (Vinograd et al., 2017).

2) The authors should comment on the identity and experience-dependent regulation of odorant receptors that are overrepresented in male mice.

We have greatly expanded on our discussion of male-enriched ORs Olfr1437 and Olfr235. We were also able to identify in vivo agonists for OSNs expressing these ORs both in our lab and in the previously published literature. Remarkably, agonists for these male-enriched ORs are musk-related odorants. As a result, we now more broadly address sexual dimorphism in the MOE rather than focusing exclusively on female-enriched populations of OSNs throughout the manuscript.

3) All supplemental figures should be referenced in the main text.

We have now directly addressed all supplemental figures in our main text.

4) The authors discuss their work in the broad context of inter-individual variability. Sexual dimorphism is not an example of inter-individual variability. The authors may want to rephrase the general context of this study.

We are not the first to suggest sexual dimorphism to be an example of inter-individual variability (Xu et al., 2016, Dulac and Kimchi, 2008).

And indeed, the over-representation of Olfr910, Olfr912, and Olfr1295 only occur when female mice are sex-separated. When female mice are housed with male mice, there is no longer sexually dimorphic expression of Olfr910, Olfr912, or Olfr1295. We therefore interpret this result as an inter-individual variability amongst female mice that never occurs amongst male mice; we interpret this result to be an example of a sexual dimorphism. Nonetheless, we are also now more conservative with the usage of the word “individuality” and use the phrase “individual variability” throughout the manuscript.

Reviewer #3:The mechanisms of activity-dependent regulation of the ORs are not quite clear to me. I recommend the following relatively straight forward experiments and analysis to strengthen the paper.1) It is not clear how BAX enables activity-dependent regulation of the three receptor types. From Figure 7B, it appears there are more ORs in males, but quantification indicates a decrease in cells expressing these receptors when compared with other figures. The authors should clarify this by clearly state the effect of BAX knockout on the number of cells. They may also examine the expression levels of olfr910/912 and olfr1295 in the RNAseq data to further quantify the change.

Understanding the effects of *Bax* in OR expression frequency is complex (as reviewers 1 and 2 also point out). We now present a comprehensive evaluation the ORs, differentially expressed between wild-type and *Bax^-/-^* mice (Figure 9—figure supplement 2). Examination of the proportional abundance of OSNs expressing *Olfr910*, *Olfr912*, and *Olfr1295* between *Bax^-/-^* and wild-type mice shows that both sexes of sex-separated *Bax^-/-^* mice resemble or are slightly greater than wild-type sex-separated males. Based on the expanded epithelial thickness in *Bax^-/-^* mice (Robinson et al., 2003) and the change in relative abundance of hundreds of ORs in *Bax^-/-^* mice, one way to interpret this result is that Bax deficiency resulted in a prolonging lifespan of OSNs expressing hundreds of ORs responding to environmental or self-generated odorants.

There are also many other interpretations to these results, and we are not partial to any of them. Therefore, we are very conservative in the implications of the results from the *Bax^-/-^* mice. Nonetheless, we believe the results are still very informative of a potential underlying mechanism, and are therefore continue to include the data. Shown in Author response image 1, Author response image 2 and Author response image 3 are the generated plots to compare OR representation frequency between wild-type and *Bax^-/-^* mice.

**Author response image 2. respfig2:** 

**Author response image 3. respfig3:** 

2) If BAX is expressed broadly, how does it control the activity-dependent survival of only a few receptor types? It is not clear how this specificity is achieved. The authors may address this issue by examining whether BAX is expressed at a higher level by neurons expressing these receptors. If not, the authors should discuss the involvement of other potential mechanisms.

We do not propose that *Bax* exerts influence only on OSN populations expressing these receptors. We now indicate this by differentially expressing *Bax^-/-^* and wild-type mice to show differential expression of many receptors. Given our manuscripts emphasis on the role of experience in OR representation, we limit our focus to the effects of Bax on OR representation in an experience-dependent manner. We also searched for genes enriched amongst OSNs expressing *Olfr910*, *Olfr912*, and *Olfr1295* from a recently published single-cell RNA-Seq dataset and found primarily activity-related genes and include this in our Discussion (Figure 8). We also now cite a recent paper showing changes in OR representation likely by changes in OSN development as an example of alternative mechanisms (van der Linden et al., 2020).